# Equivariant Latent Alignment via Flow Matching under Group Symmetries

**Sunghyun Kim** [*1]  **Jaehoon Hahm** [*2]  **Jeongwoo Shin** [1]  **Joonseok Lee** [1]

## Abstract

Geometry-aware generative models and novel view synthesis approaches have shown strong potential in visual fidelity and consistency. In parallel, equivariant representation learning has emerged as a powerful framework for constructing latent spaces where analytically known group transformations could act directly, capturing geometric structure in data and enhancing both interpretability and generalization in novel view synthesis. However, we identify that existing approaches often suffer from *latent misalignment*, a discrepancy between the intended group action and the actually required transformations in the latent space. Consequently, the learned latents often fail to consistently preserve the equivariant relations imposed by the underlying group symmetry. To address this, we propose **Residual Latent Flow**, a flow-based framework that corrects the misaligned latents, thereby improving compliance with the underlying equivariance relation. Our comprehensive experiments show that our method significantly reduces latent misalignment and improves novel view synthesis quality, under rotation groups $SO(n)$.

## 1. Introduction

Recent advances in geometry-aware generative models, such as diffusion (Karnewar et al., 2023; Yu et al., 2023; Shi et al., 2024; Anciukevičius et al., 2023) and Generative Adversarial Networks (Chan et al., 2022), have significantly enhanced visual fidelity in generation. Moreover, geometry-aware novel view synthesis (NVS) approaches (Miyato et al., 2022; Koyama et al., 2024; Miyato et al., 2024) generate realistic images of scenes or objects from previously unseen

viewpoints by leveraging geometric structure of its latent space, enabling consistent novel view synthesis and diverse computer vision applications.

In parallel, equivariant representation learning (Cohen & Welling, 2016; Falorsi et al., 2018; Dupont et al., 2020; Quessard et al., 2020) leverages intrinsic symmetries in training data to enforce structured transformation behavior in the learned representations. Ensuring the latent representations to transform predictably under group actions (*e.g.*, rotation), such models offer improved generalization and interpretability of the latent space. This provides interpretable latent transformations aligned with the structure of compact Lie groups (Finzi et al., 2020; Ruhe et al., 2023).

Formally, a mapping $\Phi$ is *equivariant* if the mapping's representations *corotate* for a group action such as rotation is applied to the data: $\rho(g)\Phi(x) = \Phi(g \circ x)$, where $\rho$ is a pre-defined group representation for the elements of the interested symmetry group $g \in G$. Intuitively, the encoder $\Phi$ is equivariant if the latent upon the group action $\rho(g)\Phi(x)$ is aligned with the corresponding latent from the transformed image $\Phi(g \circ x)$.

An encoder-based equivariant representation learning relies on a strong assumption that the encoder $\Phi$ jointly learns both how to compress the object content and the underlying symmetry group's structure in a perfectly equivariant manner. In practice, however, we discover misalignment between the analytically rotated latent $\rho(g)\Phi(x)$ and the true target latent $\Phi(g \circ x)$, even when the encoder attains faithful reconstructions.

A more fundamental limitation also arises from the aliasing of intermediate feature representations. Prior works have shown that standard convolutional and transformer-based architectures inherently introduce aliasing due to discrete sampling and non-band-limited filters, which leads to persistent equivariance errors even under idealized datasets (Karras et al., 2021; Azulay & Weiss, 2019; Rahaman et al., 2019). We succinctly refer to this issue as *latent misalignment*, which undermines the equivariance and degrades the fidelity of synthesized views under transformations.

To address this, we propose a latent correction mechanism termed **Residual Latent Flow**, which learns a transport

---

[*]Equal contribution  [1]Seoul National University, Seoul, Korea [2]University of Illinois Urbana-Champaign, Illinois, USA. Correspondence to: Joonseok Lee <joonseok@snu.ac.kr>.

*Proceedings of the 43rd International Conference on Machine Learning*, Seoul, South Korea. PMLR 306, 2026. Copyright 2026 by the author(s).

from the analytically transformed latent $\rho(g)\Phi(x)$ to its empirically encoded counterpart $\Phi(g \circ x)$ using Flow Matching (Lipman et al., 2023; Liu et al., 2023; Albergo et al., 2025; Tong et al., 2023). Rather than discarding the group-theoretic prior, such as the Wigner $D$-matrix representation of the rotation group $SO(3)$, we treat the known group action $\rho(g)$ as a first-order approximation and learn residual corrections on top of it.

Our correction framework is distinguished from conventional applications of flow matching that transport simple noise distributions to data. Here, the transport must preserve correspondence between paired latents arising from the same object under a known group action. The correction must behave consistently across different starting views, while remain flexible enough to capture object dependent deviations. Flow matching is well-suited to this setting, as it allows a flexible choice of source and target distributions and offers freedom to suffice boundary conditions required from this specific transport problem under group symmetry.

Our main contributions can be summarized as follows:

- **Identification of latent misalignment in equivariant models.** We define and analyze the phenomena of *latent misalignment* that arise from discrepancies between analytically transformed and empirically encoded representations, and show how it undermines geometric consistency.
- **Latent correction via flow model**. We propose a latent correction framework based on flow matching to resolve latent misalignment, enabling iterative and data-driven correction, under group symmetry.
- **Improvements in consistency and novel view synthesis quality.** We demonstrate that our method effectively improves latent alignment and enhances novel view synthesis across multiple datasets that inherently admit rotational symmetries, such as $SO(2)$ and $SO(3)$.

## 2. Background

### 2.1. Equivariance under Group Symmetry

Equivariant representation learning (ERL) leverages symmetry properties inherent in data to capture latent structural relationships. Formally, a mapping $\Phi$ is equivariant with respect to a group $G$ if it satisfies the following relation:

$$\Phi(g \circ x) = \rho(g)\Phi(x), \quad (1)$$

$\forall x \in X, \forall g \in G$, where $\circ : G \times X \to X$ is the group action on the set of data $X$, $\rho : G \to GL(n, \mathbb{R})$ is the group

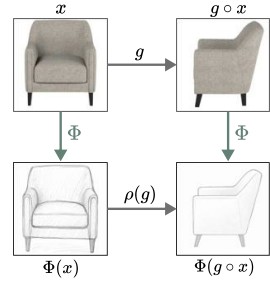

*Figure 1.* Illustration of equivariance relation.

representation of group $G$. Then, a complementary decoder $\Psi$ targets to recover the original input from the latent representation: $\Psi(\Phi(x)) = x, \ \forall x \in X$.

In practice, ERL aims to ensure that latent features transform consistently with the underlying group action. To achieve this, $\Phi$ and $\Psi$ are trained with the equivariance loss, which explicitly penalizes deviations from the equivariance relation, and a reconstruction loss, which encourages preservation of sufficient information for recovery:

$$\mathcal{L}_{\text{ERL}} = \underbrace{\mathbb{E}_{x,g}[\|\Phi(g \circ x) - \rho(g)\Phi(x)\|_2^2]}_{\text{Equivariance Loss}}$$

$$+ \underbrace{\mathbb{E}_{x,g}[\|g \circ x - \Psi(\rho(g)\Phi(x))\|_2^2]}_{\text{Decoder Loss}} \quad (2)$$

### 2.2. Special Orthogonal Groups

In this work, we focus on the datasets that explicitly include controlled group action, specifically rotation, *e.g.*, turntable scans or object-centric synthetic renders. These are collected for novel view synthesis (NVS) as multi-view captures of a scene, which can be naturally modeled by the special orthogonal groups $SO(n) = \{\mathbf{R} \in GL(n, \mathbb{R}) \mid \mathbf{R}^\top \mathbf{R} = \mathbf{R}\mathbf{R}^\top = I, \det(\mathbf{R}) = 1\}$.

$SO(3)$ is the spherical orthogonal group which consists of rotations in three-dimensional space. The group elements can be parametrized with three Euler angles $\alpha, \beta, \gamma$, as $R(\alpha, \beta, \gamma) = e^{-i\alpha J_z}e^{-i\beta J_y}e^{-i\gamma J_z}$, where $J_x, J_y, J_z$ are the generators (angular momentum operators) of the Lie algebra $\mathfrak{so}(3) = \{\mathbf{A} \in GL(3, \mathbb{R}) \mid \mathbf{A}^\top = -\mathbf{A}\}$ of $SO(3)$. In parallel, there exists Wigner $D$-matrix representation $D^{(\ell)} : SO(3) \to GL(2\ell + 1, \mathbb{C})$ of degree $\ell$ that maps the rotation group element to a $(2\ell + 1) \times (2\ell + 1)$ matrix: $D_{m,n}^{(\ell)} = e^{-im\alpha}d_{m,n}^{(\ell)}(\beta)e^{-in\gamma}$, where $d_{m,n}^{(\ell)}(\beta)$ is the real-valued Wigner small-$d$ matrix depending on the polar angle $\beta$ and $m, n \in \{-\ell, \cdots, \ell\}$ index the basis states of the degree-$\ell$ irreducible representation.

$SO(2)$ is the subgroup of $SO(3)$ consisting of rotations about the $z$-axis. The irreducible unitary representations of $SO(2)$ are all one-dimensional characters, indexed by an integer frequency $m \in \mathbb{Z}$: $\rho_m(\theta) = e^{im\theta}$. These characters arise naturally by restricting the Wigner $D$-matrices of $SO(3)$ (*i.e.*, $\beta = \gamma = 0$ and $\alpha = \theta$). Refer to Section C for more details.

### 2.3. Flow Matching

Flow matching is a generative modeling framework that seeks to transform samples from a prior distribution $p_0$ to a target distribution $q$ through a continuous-time flow induced by an ordinary differential equation (ODE): $\frac{d}{dt}\psi_t(z) = v_t(\psi_t(z))$, where flow $\psi : (t, z) \mapsto \psi_t(z) = z_t$ is a time-dependent diffeomorphism that pushforwards $p_0$ to $q$ and

velocity field $v : (t, z) \mapsto v_t(z)$ is a solution to the ODE. Then, given a density $p_0$ at $t = 0$, the probability path $p_t : \mathbb{R}^d \to \mathbb{R}$ can be identified as the pushforward of $p_0$ under flow, $p_t = \psi_t^\# p_0$. Here, $z_t \in \mathbb{R}^d$ denotes a sample at time $t$ along the trajectory that connects the initial distribution $p_0$ to the target distribution $p_1$, and $t \in [0, 1]$ is an artificial time variable that parameterizes the flow. At $t = 0$, the sample $z_t$ is drawn from $p_0$, and as $t \to 1$, the trajectory governed by the learned vector field transports $z_t$ toward the target distribution $p_1$. The goal is to learn a velocity field $v(z_t, t)$ such that integrating the ODE yields a distributional mapping from $p_0$ to $p_1$ over time.

Since the true marginal $v_t$ is intractable, Lipman et al. (2023); Liu et al. (2023); Albergo et al. (2025) introduce a simulation-free training framework using conditional flow matching loss:

$$\mathcal{L}_{\text{CFM}}(\theta) = \mathbb{E}_{t, \varepsilon, z_t} \left[ \|v_\theta(t, z_t) - v_t(z_t|\varepsilon)\|_2^2 \right], \quad (3)$$

where $t \sim U[0, 1]$, $\varepsilon \sim q(\cdot)$, $z_t \sim p_t(\cdot|c)$.

**Equivariant Representation Learning.** We mainly follow Neural Fourier Transform (NFT) (Koyama et al., 2024; Miyato et al., 2022) for our analysis and experiments. Specifically, NFT considers a basis transform $P$ that block diagonalizes the group representation $\tilde\rho$ of a given group $G$, facilitating the decomposition into $n$ irreducible components:

$$\rho(g) = \bigoplus_{i=1}^n \rho_i(g), \qquad \rho(g) = P\tilde\rho(g)P^{-1}, \quad (4)$$

where $\rho$ is the block-diagonal representation after applying similarity transformation to $\tilde\rho$. Then $\rho$ can be identified as a direct sum of irreducible representations $\rho_i$. The $\ell$-th irreducible block of the representation can be identified as the Wigner $D$-matrix of degree-$\ell$, i.e., $\rho_\ell(g) = D^{(\ell)}(g)$. The NFT framework provides interpretable block components via block-level equivariant learning, offering a fine-grained evaluation.

After representing the group elements $g \in G$ as a direct sum of degree-$\ell$ Wigner $D$-matrix, we are interested in training an autoencoder that learns to map data to a latent representation such that *corotate* for a group action: $\rho(g)\Phi(x) = \Phi(g \circ x)$. Hence, given the group representation $\rho : G \to \oplus_{\ell=0}^L GL(\dim(\rho_\ell), \mathbb{R})$, where $L$ is the maximum degree of the representations, we consider training an encoder $\Phi : \mathbb{R}^{H \times W} \to \mathbb{R}^{C \times N_G}$ and a decoder $\Psi : \mathbb{R}^{C \times N_G} \to \mathbb{R}^{H \times W}$ by minimizing the ERL loss (Equation (2)). Here, $H, W$ denote the dimension of input image, $C$ is the latent representation's channel size, and $N_G$ is the sum of dimensions of all degree-$\ell$ representations of group $G$. For example, in SO(2), $N_{\text{SO(2)}} = 1 + 2L$ because the representation comprise one scalar block for trivial representation ($\ell = 0$) and $L$ $2 \times 2$ rotation blocks for other non-zero

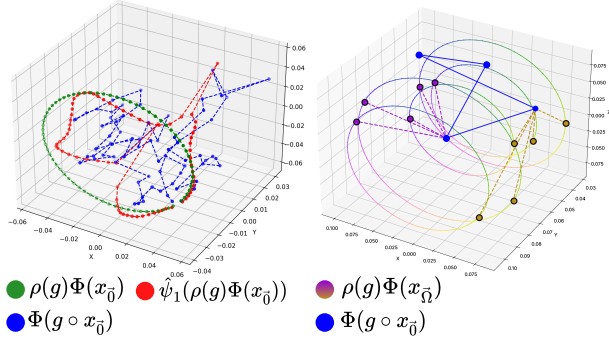

● $\rho(g)\Phi(x_{\vec{0}})$  ● $\hat\psi_1(\rho(g)\Phi(x_{\vec{0}}))$  ● $\rho(g)\Phi(x_{\vec\Omega})$
● $\Phi(g \circ x_{\vec{0}})$  ● $\Phi(g \circ x_{\vec{0}})$

*Figure 2.* *Left:* **Visualization of latent trajectories under** SO(3). Green and blue dots depict analytically transformed latents $\rho(g)\Phi(x_{\vec{0}})$ and encoder-derived latents $\Phi(g \circ x_{\vec{0}})$, respectively. Red dots depict corrected latents by our method. We utilize degree-1 representation of SO(3) rotation for visualization. *Right:* **Motivation to use flow matching.** Each cyclic trajectory, colored with a smooth cyclic colormap, corresponds to the latent orbit of different initial views, $\text{Orb}(\Phi(x_{\vec{\Omega}_i})) := \{\rho(g)\Phi(x_{\vec{\Omega}_i}) \mid g \in G\}$. The blue target latents, $\Phi(g_j \circ x_{\vec{0}})$, and their corresponding source latents (other colors), $\rho(g_j g_{\vec{\Omega}_i}^{-1})\Phi(x_{\vec{\Omega}_i})$ are misaligned. We address this as a distribution transport problem, utilizing flow matching.

degree representation. In SO(3), $N_{\text{SO(3)}} = \sum_{\ell=0}^L (2\ell + 1)$ which is the concatenated dimensionality of all degree-$\ell$ vectors of SO(3).

## 3. Method

### 3.1. Identification of Latent Misalignment in ERL

As illustrated in Figure 2 (left), the actual latent paths (blue) learned by the encoder form irregular, jagged trajectories, far from the analytically derived paths (green). In other words, for the same object, analytically rotated $\rho(g_{\Delta\theta})\Phi(x_\theta)$ and true encoding $\Phi(x_{\theta + \Delta\theta})$ diverges. We further observe that this discrepancy grows with rotation magnitude $\Delta\theta$, implying accumulation of the error (See Figure 4). See Section K for more discussions about the cause of latent misalignment.

This highlights the need for a *correction mechanism* for refining the learned equivariant latent representations. To this end, we aim to mitigate this gap in reality by introducing a principled transportation mechanism while preserving the group-theoretic foundation.

### 3.2. Residual Latent Flow for Latent Correction

Based on our observations of latent misalignments in ERL, we propose **Residual Latent Flow** (RLF), a method for latent correction based on flow matching (Lipman et al., 2023; Liu et al., 2023; Albergo et al., 2025). Specifically, we employ the analytically transformed latent $\rho(g)\Phi(x)$ as a first-order approximation and let continuous flow to transport the latent to its corresponding target $\Phi(g \circ x)$.

**Problem Formulation.** Let $(x, g) \sim q$ be a data pair sam-

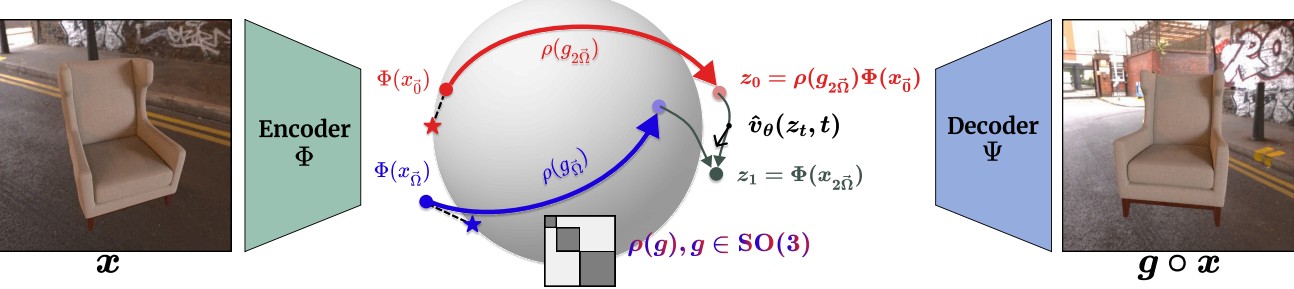

*Figure 3.* **Illustration of our Residual Latent Flow.** Standard encoder-based equivariant representation learning frameworks suffer from *latent misalignment*, where the learned latent codes do not align with the intended equivariant structure, i.e. $\rho(g)\Phi(x) \neq \Phi(g \circ x)$. In practice, real latent trajectories (circles) deviate from the ideal ones (stars), resulting in inconsistent endpoints. Two images obtained by viewing single object from two different angles are depicted as $x_{\vec{0}}$ and $x_{\vec{\Omega}}$. Our method introduces a flow based latent correction step that explicitly realigns latents. The correction step explicitly enforces $\rho(g)\Phi(x) \approx \Phi(g \circ x)$, which restores consistency in the latent space and improves visual fidelity in novel view synthesis.

pled from a dataset, where $x$ is an image and $g \in G$ is a known group transformation. Let us denote the latent representation of the original image by $\Phi(x)$. Then, the analytic group transformation applied to latent and the latent of the transformed image can be denoted as

$$z_0 := \rho(g)\Phi(x), \qquad z_1 := \Phi(g \circ x), \qquad (5)$$

respectively. We refer to the distributions of $z_0$ and $z_1$ as the source distribution $p_0$ and target distribution $p_1$, respectively. As shown in Section 3.1, $z_0$ and $z_1$ are not precisely aligned in practice due to the imperfect encoder and data variability. To correct this misalignment, we connect the $z_0$ to its corresponding $z_1$ via training an adequate flow $\psi : [0,1] \times \mathbb{R}^d \to \mathbb{R}^d$.

**Why Flow Matching for Latent Correction?** We view the task of correcting equivariant latents as a distribution transport problem, as we are trying to match $\rho(gh^{-1})\Phi(h \circ x_0)$ with $\Phi(g \circ x_0)$ for every $g, h \in G$; *i.e.*, there exists multiple source points $\{\rho(gh^{-1})\Phi(h \circ x_0)\}_{h \in G}$ which needs to coincide with a single target point, for every $g \in G$.

Moreover, the latents trained with ERL loss has an inevitable misalignment, due to contraction caused by the imperfectly optimized encoder. The issue arises when there exist distinct inputs $x \neq x'$ such that $\Phi(x) = \Phi(x')$, i.e., the encoder maps different images to the same latent representation. In this case, after applying the group action $g$, we obtain two valid targets $\Phi(g \circ x)$ and $\Phi(g \circ x')$ for the same source $\rho(g)\Phi(x) = \rho(g)\Phi(x')$, resulting in a multi-modal target distribution. This is not a pathological case but naturally occurs in realistic settings where we are given an imperfectly trained encoder. In natural image data, different scenes or objects can share similar latent representations due to limited encoder capacity and invariances. Within the ERL framework, this is further exacerbated by imperfect optimization of the equivariance objective, discretization, and

architectural biases, which prevent the encoder from learning fully injective and perfectly equivariant mappings.

Our approach is to explicitly address this regime by learning a generative model to transport between distributions rather than learning a deterministic point-to-point mapping in order to model a multi-modal target distribution. Unlike direct regression, which might be a natural first attempt to resolve this problem, flow matching provides an iterative mapping that can flexibly adapt to arbitrary source and target distributions. Diffusion-based approach may also come to mind as an alternative due to their iterative refinement procedure. However, standard diffusion formulations are typically defined using Gaussian noise priors, whereas flow-based transport offers flexibility in choosing an arbitrary prior. As visualized in Figure 2, our method aims to transport the distribution of purple and yellow source latents to the target distribution of blue latents.

**Latent Correction by Flow Matching.** To achieve this, we need to design a probability path that starts from $z_0$ and arrives at $z_1$. Considering this boundary condition, we derive the marginal probability path by marginalization:

$$p_t(z_t) = \int p_t(z_t|z_0, z_1)\,\pi_{0,1}(z_0, z_1)\,\mathrm{d}z_0\,\mathrm{d}z_1, \quad (6)$$

where $z_0 \sim p_0$, $z_1 \sim p_1$ and $\pi_{0,1}$ is the joint distribution of the variables $z_0, z_1$. However, unlike the previous flow matching frameworks with arbitrary priors (Liu et al., 2023; Albergo et al., 2025), we further need to consider the specific characteristics of this problem; that is, the flow must transport the given $z_0$ to its *corresponding* $z_1$ (as defined in Equation (5)), which is more complex than the ordinary flow matching that aims to learn transportation between two distributions in marginal level via *independent* sampling of $z_0$ and $z_1$. For example, the flow must map the latent of a specific sofa to the latent of the *same* sofa rotated by a given an-

gle, not to the rotated latent of a random object. As a result, we cannot sample $z_0$ and $z_1$ independently to construct the conditional path as usual, *i.e.*, $\pi_{0,1}(z_0, z_1) \neq p_0(z_0)p_1(z_1)$ in this problem.

A simple solution, without introducing extra overhead (*e.g.*, object-conditional embeddings), is naturally following the definitions of $z_0$ and $z_1$ in Equation (5). Viewing the samples $z_0 \sim p_0(\cdot|x,g)$ and $z_1 \sim p_1(\cdot|x,g)$ are from conditional distributions, it is natural to formulate the conditional joint distribution as $\pi_{0,1}(z_0, z_1|x,g) = p_0(z_0|x,g)p_1(z_1|x,g)$. This conditional probability path supervises the flow model via *informative* trajectory, which is different from the previous generative flows with independent coupling. Plugging this into Equation (6) with additional marginalization over distribution of training data $(x,g) \sim q$ leads to:

$$p_t(z_t) = \int p_t(z_t|z_0, z_1)\, p_0(z_0|x,g)\, p_1(z_1|x,g)\, q(x,g)\, \mathrm{d}\Omega, \tag{7}$$

where $\mathrm{d}\Omega := \mathrm{d}z_0\, \mathrm{d}z_1\, \mathrm{d}x\, \mathrm{d}g$. Although Equation (7) introduces four marginalizations, the conditional probability of $z_0$ and $z_1$ collapse to Dirac measures, since both are deterministically obtained from $(x,g)$ via the map $\Phi$ (Equation (5)), *i.e.*, $p_0(\cdot \mid x,g) = \delta_{\rho(g)\Phi(x)}(\cdot)$ and $p_1(\cdot \mid x,g) = \delta_{\Phi(g \circ x)}(\cdot)$. Hence, Equation (7) reduces to

$$p_t(z_t) = \int p_t\big(z_t|\rho(g)\Phi(x),\, \Phi(g \circ x)\big)\, q(x,g)\, \mathrm{d}x\, \mathrm{d}g. \tag{8}$$

Then, we instantiate the conditional probability path as a linear interpolation following Liu et al. (2023) between $z_0 = \rho(g)\Phi(x)$ and $z_1 = \Phi(g \circ x)$:

$$p_t(z_t \mid z_0, z_1) = \mathcal{N}(z_t|(1-t)z_0 + tz_1, \sigma^2 I). \tag{9}$$

where $\sigma$ controls the strength of stochasticity. For $\sigma = 0$, conditional velocity field becomes constant over time $v_t(z_t|z_0, z_1) = z_1 - z_0$, and our Residual Latent Flow (RLF) training objective is given by:

$$\mathcal{L}_{\mathrm{RLF}}(\theta) = \mathbb{E}_{x,g,t,z_t}\left[\|v_\theta(z_t, t) - (z_1 - z_0)\|_2^2\right], \tag{10}$$

where $(x,g) \sim q$, $t \sim \mathcal{U}[0,1]$, $z_t \sim p_t(\cdot|z_0, z_1)$ and $z_0 = \rho(g)\Phi(x)$, $z_1 = \Phi(g \circ x)$. Upon convergence of the loss, flow $\hat{\psi}_1$ can be obtained by integrating the learned velocity field predictor $v_\theta$ from time 0 to 1, to transport $z_0$ to its $z_1$, *i.e.*, $\hat{\psi}_1(z_0) = \hat{z}_1 = z_0 + \int_0^1 v_\theta(z_\tau, \tau)\mathrm{d}\tau$.

### 3.3. Training

We provide full training procedure of our method, which is also summarized in Algorithm 1.

**Baseline Autoencoder Training.** Our baseline implementation follows the original NFT setup (Koyama et al., 2024).

We utilize ViT (Dosovitskiy et al., 2021) for both encoder $\Phi$ and decoder $\Psi$. First, we train the autoencoder with the ERL loss (Equation (2)) to get the latent representations.

**RLF Training.** Then, we freeze the encoder and train our flow model with our RLF loss (Equation (10)) to correct the latents obtained from using the pre-trained encoder. We also attempted end-to-end training, but the process was unstable. We hypothesize that this instability stems from the latent distribution drifting as the autoencoder and flow model simultaneously updates.

**Decoder Fine-tuning.** After latent correction, the decoder $\Psi$ consumes flow-corrected latents, so we fine-tune $\Psi$ directly on that input distribution to remove the distribution shift between training and use. We update only $\Psi$ by freezing the encoder $\Phi$ and the flow model $\hat{\psi}_1$ and minimizing

$$\mathcal{L}_{\mathrm{fine\text{-}tune}} = \mathbb{E}[\|g \circ x - \Psi(\mathrm{sg}(\hat{\psi}_1)(\rho(g)\mathrm{sg}(\Phi)(x)))\|_2^2]. \tag{11}$$

This objective adapts the decoder to its flow-corrected latent input distribution, aligning $\Psi$ to the corrected latent manifold while preserving supervision via the ground-truth image $g \circ x$. Freezing the encoder $\Phi$ and the flow model $\hat{\psi}_1$ prevents modifications that would undo the learned equivariance or the flow correction. We find that in practice, this yields more faithful reconstructions and better synthesis quality for both baseline and flow-based models.

## 4. Experiments

### 4.1. Datasets

We evaluate our method on six datasets that collectively cover geometric group transformations ($\mathrm{SO}(2), \mathrm{SO}(3)$), structured appearance variations for both synthetic and real-image settings.

Four datasets are synthetically generated, allowing precise control over group actions and appearance factors. **ABO-Material** (Collins et al., 2022) and **ModelNet10-SO(3)** (Liao et al., 2019) have the $\mathrm{SO}(3)$ symmetry; in ABO-Material, the object and its background co-rotate under viewpoint changes, whereas in ModelNet10-SO(3), the object rotates without background. **ComplexBRDFs** (Greff et al., 2022) has $\mathrm{SO}(2)$ symmetry, where the objects rotate only by a fixed axis. **ABO-Material Day-to-Night** is a novel $\mathrm{SO}(2)$-style appearance variant derived from the ABO-Material (Collins et al., 2022), in which the object geometry and viewpoint are fixed while the illumination is systematically varied to induce day-to-night-like changes in shadows and color tones.

To assess robustness beyond synthetic renderings, we additionally evaluate on two real-image datasets. **RotatedMNIST** (Deng, 2012) is an $\mathrm{SO}(2)$ in-plane rotation benchmark constructed from the MNIST dataset (Deng, 2012),

where each digit image is transformed by an in-plane $SO(2)$ rotation. **SmallNORB** (LeCun et al., 2004) is a real-image dataset featuring object-level $SO(3)$ rotations under varying lighting conditions and viewpoints. More details are provided in Section F.1.

### 4.2. Implementation Details

**Training Setup.** For the baseline autoencoder, we largely follow the original NFT training configurations (Koyama et al., 2024). Specifically, ABO-Material, ComplexBRDFs, and ModelNet10-$SO(3)$ use the default NFT setup. Small-NORB is trained using the same configuration as ABO-Material. For ABO-Material Day-to-Night and RotatedM-NIST, we adopt the configuration used for ComplexBRDFs.

For datasets with smaller spatial resolution and grayscale inputs (RotatedMNIST and SmallNORB), we modify the ViT-based encoder-decoder architecture by reducing the image size and setting the input channel dimension to one.

Flow-based models are trained for 300 epochs using AdamW with a batch size of 128, learning rate $10^{-4}$, and weight decay 0.05. After that, Decoder fine-tuning is performed for 10% of the original training epochs with the learning rate reduced by a factor of 10.

**Architectures.** For $SO(2)$, we parameterize the flow model using a Transformer-based architecture with 4 self-attention layers, 8 heads, and latent channel dimensions of 128 (0.8M parameters) or 256 (2.1M parameters).

For $SO(3)$, we employ a U-Net (Ronneberger et al., 2015)-based architecture for the flow model. The $SO(3)$ latent representation is formed by concatenating Wigner $D$-matrix blocks up to degree $L$, whose total dimensionality satisfies $\sum_{l=0}^{L}(2l+1) = (L+1)^2$. This naturally admits a square-grid reshaping, allowing the flow to be implemented using convolutional layers in a computationally efficient manner.

### 4.3. Evaluation Metrics

Let $x \in \mathbb{R}^{H \times W \times 3}$ be an input image and $g \in G$ a group element. We compare the performance of the following two models:

$$\hat{x}_{\text{base}}(g) := \Psi(\rho(g)\Phi(x)),$$
$$\hat{x}_{\text{ours}}(g) := \Psi\left(\hat{\psi}_1(\rho(g)\Phi(x))\right),$$

where our Residual Latent Flow transports the analytic latent $\rho(g)\Phi(x)$ toward the target latent $\Phi(g \circ x)$. We use the following metrics to evaluate performance.

**Prediction Error.** We report the mean L2 distance between the synthesized image and the ground-truth transformed image in the pixel space $\|\hat{x}(g) - g \circ x\|_2^2$, where $\hat{x}(g) \in \{\hat{x}_{\text{base}}(g), \hat{x}_{\text{ours}}(g)\}$ and the ground-truth transformed image

is denoted as $g \circ x$.

**LPIPS.** We report the Learned Perceptual Image Patch Similarity metric (Zhang et al., 2018) to measure perceptual similarity between the synthesized image $\hat{x}(g)$ and the ground-truth transformed image $g \circ x$. LPIPS evaluates perceptual differences using deep features extracted from a pretrained network, and better correlates with human visual judgment.

**Peak Signal-to-Noise Ratio (PSNR).** Computed between the predicted image $\hat{x}(g) \in \{\hat{x}_{\text{base}}(g), \hat{x}_{\text{ours}}(g)\}$ and the ground-truth transformed image $g \circ x$, it quantifies the reconstruction fidelity in the pixel space using a logarithmic scale.

**Latent Error.** Similarly, we report the L2 distance between the predicted rotated latent and the latent of ground-truth transformed image in the latent space $\left\|\Psi^{-1}(\hat{x}(g)) - \Phi(g \circ x)\right\|_2^2$, where $\hat{x}(g) \in \{\hat{x}_{\text{base}}(g), \hat{x}_{\text{ours}}(g)\}$ and the latent of ground-truth transformed image is denoted as $\Phi(g \circ x)$.

**Angle Error.** To measure the strength of violation of latent equivariance, we compare the predicted latent $\Phi(g \circ x)$ with the analytically rotated source latent $\rho(g)\Phi(x)$. For $SO(2)$, we estimate the relative rotation angle that best aligns the corresponding representation blocks and report an angular discrepancy aggregated across degrees. For $SO(3)$, we estimate the relative rotation using the degree-1 block via a Wahba-type alignment and measure its deviation from the identity rotation. Higher-degree $SO(3)$ representations can be evaluated using latent errors. Full mathematical details are provided in Section E.

### 4.4. Comparison on Novel View Synthesis

**In-plane Rotation Synthesis.** First, we evaluate whether our method is effective for in-plane rotation synthesis, a special case of NVS where the rotation axis is parallel to the viewing direction, *e.g.*, digits rotating within the same plane in RotatedMNIST. In Table 2, we show comparison between Spatial-VAE (Bepler et al., 2019), GIAE (Shakerinava et al., 2022), LGA (Jin et al., 2024), NFT (Koyama et al., 2024), and ours. In Figure 5, we provide qualitative examples. This result corroborates that our latent correction method is effective and outperforms the existing baselines.

**Out-of-plane Rotation Synthesis.** Out-of-plane rotation synthesis is another form of novel view synthesis, but in a more challenging setting: the object rotates around an axis that is not parallel (typically orthogonal) to the viewing direction. In this case, the model has to infer the appearance of occluded parts, making the task significantly harder.

In Table 1, we show comparison across various datasets that has $SO(2)$ or $SO(3)$ group symmetry. We evaluate performance under two conditions: *(i)* out-of-distribution

*Table 1.* **Comparison on various novel view synthesis tasks.** We report prediction error, LPIPS, PSNR, and latent error to measure the reconstruction quality and angle error to measure the violation strength of latent equivariance. Base indicates NFT (Koyama et al., 2024). These values are averaged over the entire test or OOD split for each dataset. Refer to Section F.1 for details.

| GROUP | DATASET | METHOD | PRED ERR.↓ | LPIPS↓ | PSNR↑ | LATENT ERR.↓ | ANGLE ERR.↓ |
|---|---|---|---|---|---|---|---|
| SO(3) | ABO | Base | 0.0667 ±0.0004 | 0.4434 ±0.0002 | 11.85 ±0.01 | $7.3 \times 10^{-4}$ ±1.3×10⁻⁶ | 0.0079 ±0.0001 |
| | | **Ours** | **0.0565** ±0.0004 | **0.4267** ±0.0002 | **12.57** ±0.01 | $\mathbf{1.9 \times 10^{-4}}$ ±5.1×10⁻⁷ | **0.0010** ±0.0001 |
| | ABO (OOD) | Base | 0.0641 ±0.0008 | 0.4405 ±0.0003 | 12.14 ±0.01 | $8.1 \times 10^{-4}$ ±1.7×10⁻⁶ | 0.0088 ±0.0002 |
| | | **Ours** | **0.0564** ±0.0008 | **0.4251** ±0.0003 | **12.73** ±0.01 | $\mathbf{2.1 \times 10^{-4}}$ ±7.6×10⁻⁷ | **0.0012** ±0.0001 |
| | ModelNet10-SO(3) (OOD) | Base | 0.1079 ±0.0025 | 0.1176 ±0.0005 | 10.09 ±0.03 | $7.2 \times 10^{-4}$ ±7.1×10⁻⁶ | 0.1746 ±0.0064 |
| | | **Ours** | **0.1018** ±0.0026 | **0.1084** ±0.0005 | **10.43** ±0.03 | $\mathbf{4.1 \times 10^{-4}}$ ±7.4×10⁻⁶ | **0.0430** ±0.0018 |
| | SmallNORB (OOD) | Base | 0.0052 ±0.0001 | 0.2729 ±0.0002 | 23.13 ±0.02 | $7.9 \times 10^{-5}$ ±2.7×10⁻⁷ | 0.2429 ±0.0023 |
| | | **Ours** | **0.0050** ±0.0001 | **0.2473** ±0.0002 | **23.28** ±0.02 | $\mathbf{4.8 \times 10^{-5}}$ ±2.6×10⁻⁷ | **0.0728** ±0.0012 |
| SO(2) | ABO Day-to-Night | Base | 0.0056 ±0.0001 | 0.2151 ±0.0012 | 22.73 ±0.05 | $1.9 \times 10^{-4}$ ±3.0×10⁻⁶ | 0.0456 ±0.0006 |
| | | **Ours** | **0.0039** ±0.0001 | **0.1973** ±0.0011 | **24.32** ±0.05 | $\mathbf{1.5 \times 10^{-4}}$ ±2.7×10⁻⁶ | **0.0163** ±0.0003 |
| | ABO Day-to-Night (OOD) | Base | 0.0079 ±0.0005 | 0.2179 ±0.0016 | 21.80 ±0.08 | $3.4 \times 10^{-4}$ ±6.6×10⁻⁶ | 0.0776 ±0.0014 |
| | | **Ours** | **0.0065** ±0.0005 | **0.1998** ±0.0015 | **22.84** ±0.08 | $\mathbf{2.6 \times 10^{-4}}$ ±6.0×10⁻⁶ | **0.0269** ±0.0009 |
| | ComplexBRDFs | Base | 0.0382 ±0.0009 | 0.3506 ±0.0002 | 16.04 ±0.02 | $8.9 \times 10^{-4}$ ±1.3×10⁻⁵ | 0.0556 ±0.0003 |
| | | **Ours** | **0.0296** ±0.0008 | **0.3266** ±0.0002 | **17.38** ±0.02 | $\mathbf{6.7 \times 10^{-4}}$ ±1.3×10⁻⁵ | **0.0172** ±0.0004 |
| | ComplexBRDFs (OOD) | Base | 0.0480 ±0.0026 | 0.3522 ±0.0002 | 17.71 ±0.02 | $1.4 \times 10^{-3}$ ±3.8×10⁻⁵ | 0.0859 ±0.0009 |
| | | **Ours** | **0.0404** ±0.0023 | **0.3285** ±0.0002 | **19.19** ±0.03 | $\mathbf{1.1 \times 10^{-3}}$ ±3.8×10⁻⁵ | **0.0271** ±0.0010 |
| | RotatedMNIST | Base | 0.0016 ±0.0000 | 0.0035 ±0.0000 | 28.21 ±0.006 | $3.9 \times 10^{-5}$ ±1.4×10⁻⁷ | 0.0032 ±0.0000 |
| | | **Ours** | **0.0013** ±0.0000 | **0.0030** ±0.0000 | **29.02** ±0.065 | $\mathbf{3.7 \times 10^{-5}}$ ±1.4×10⁻⁷ | **0.0030** ±0.0000 |
| | RotatedMNIST (OOD) | Base | 0.0016 ±0.0000 | 0.0040 ±0.0000 | 28.17 ±0.008 | $3.9 \times 10^{-5}$ ±1.8×10⁻⁷ | 0.0032 ±0.0000 |
| | | **Ours** | **0.0013** ±0.0000 | **0.0033** ±0.0000 | **28.99** ±0.009 | $\mathbf{3.7 \times 10^{-5}}$ ±1.8×10⁻⁷ | **0.0030** ±0.0000 |

*Table 2.* **Comparison on in-plane rotation NVS using Rotat-edMNIST** (SO(2)).

| Method | Pred Err. ↓ | LPIPS ↓ | PSNR ↑ | SSIM ↑ |
|---|---|---|---|---|
| SpatialVAE | 0.0373 | 0.1561 | 15.59 | 0.6664 |
| GIAE | 0.0146 | 0.1000 | 19.88 | 0.8289 |
| LGA | 0.0049 | 0.0385 | 24.13 | 0.9427 |
| NFT | 0.0016 | 0.0035 | 28.21 | 0.9953 |
| **Ours** | **0.0013** | **0.0030** | **29.02** | **0.9961** |

*Table 3.* **Comparison on out-of-plane rotation NVS using Small-NORB** (SO(3)).

| Method | Pred Err. ↓ | LPIPS ↓ | PSNR ↑ | SSIM ↑ |
|---|---|---|---|---|
| LGA | 0.0174 | 0.270 | 17.59 | 0.785 |
| ENR | 0.0156 | 0.262 | 18.07 | 0.800 |
| NFT | 0.0052 | 0.272 | 23.13 | **0.811** |
| **Ours** | **0.0050** | **0.247** | **23.28** | 0.802 |

(OOD) objects, which were never seen during training, and *(ii)* in-distribution objects, which were seen during training but presented at unseen test-time angles. In Table 3, we compare evaluation between LGA (Jin et al., 2024), ENR (Dupont et al., 2020), NFT (Koyama et al., 2024), and ours on SmallNORB (OOD) dataset. On ModelNet10-SO(3) and SmallNORB dataset, we evaluate only on unseen objects (OOD) as no dedicated test split is available. Our method shows consistent improvements in both unseen objects and unseen angles, strongly indicating that our method

effectively corrects the misaligned latents which enhances the fidelity of decoded images.

Additionally, as shown in Figure 4, our method achieves stable improvements in all evaluation metrics regardless of the specific rotation angle, indicating that our loss objective Equation (10) is effectively designed for jointly coupled boundary distributions. Specifically, two main key components of our objective are: *(i)* $\psi_1$ transport given source point to *corresponding* target point, not to the arbitrary point of the marginal target distribution, and *(ii)* each target point should be mapped from various source points with arbitrary rotation angles. Improved performance over various datasets (Table 1) shows the effectiveness of the first component, while the improvement over all rotation angles (Figure 4) validates the second claim. Note that, the larger model (2.1M) yields the strongest performance, but even the smaller model (0.8M) consistently outperforms the baseline.

Moreover, we visualize the qualitative examples in Figure 6. The rotation angle increases from left to right in each example. We emphasize that this demonstrates generalization to unseen view angles, where test viewpoints are sampled outside the training range, which was focused as the primary objective of the work. While in-distribution viewpoints naturally yield stronger visual fidelity, our method maintains consistent geometric structure and appearance under out-of-distribution viewpoint changes. In Figure 7, we include results on ABO-material, which features high-frequency

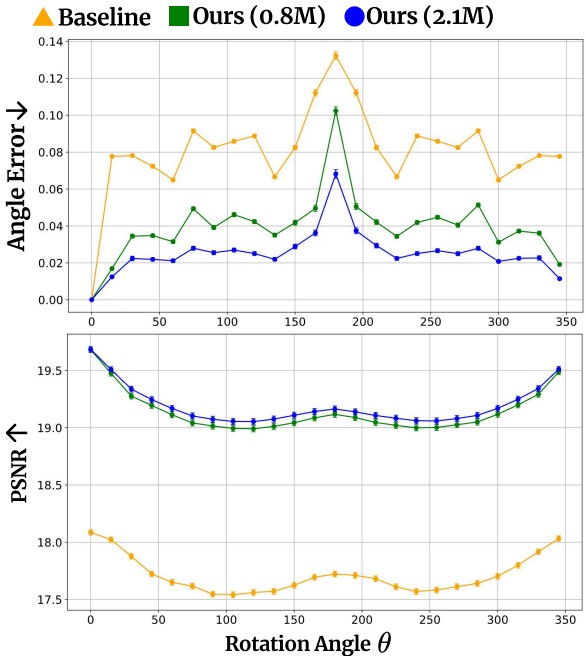

*Figure 4.* **Evaluation on** $\mathrm{SO}(2)$ **dataset (ComplexBRDFs (OOD)) across angular displacements**. The results are shown for two RLF models with 0.8M and 2.1M parameters. Rotation angle $\theta$ denotes the angular displacement (degrees) applied to every object. *Top:* Angle error as a function of rotation angle. *Bottom:* PSNR. These are evaluated on 57,360 pairs in ComplexBRDFs OOD set.

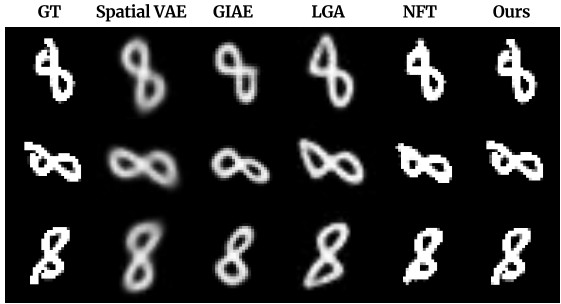

*Figure 5.* Qualitative comparison on in-plane rotation NVS ($\mathrm{SO}(2)$) on RotatedMNIST.

textures and moderately high image resolution (224×224), to support the robustness and generation quality of our approach. In Figure 8, we demonstrate examples on NVS based on $\mathrm{SO}(2)$ rotations. More qualitative examples can be found in Section L.

## 5. Related Work

**Group Symmetry-aware Models.** Equivariant neural networks and Lie group-informed models (Falorsi et al., 2018; Quessard et al., 2020; Shakerinava et al., 2022; Hayashi et al., 2025; Bertolini et al., 2025) explicitly incorporate symmetry constraints into neural architectures. Notable studies include group-equivariant convolutional networks (Cohen & Welling, 2016), homeomorphic variational auto-encoders

leveraging Lie group symmetries (Falorsi et al., 2018), and other frameworks employing group structures to enhance interpretability and robustness, including steerable and topographic parameterizations (Bökman et al., 2024; Keller & Welling, 2021), symmetry discovery and transformation-aware representations (Hinton et al., 2011; Cohen & Welling, 2015; Park et al., 2022; Miyato et al., 2022), and equivariant latent modeling (Dupont et al., 2020; Song et al., 2023b).

**Geometry-aware Diffusion and Flow Matching.** Diffusion and Flow matching (Lipman et al., 2023; Liu et al., 2023; Albergo et al., 2025) integrated with geometric modeling is an emerging area of research. These methods improve controllability and interpretability of the generative model (Hahm et al., 2024). Recent work demonstrates Diffusion and Flow matching on general geometries (Chen & Lipman, 2024; Sherry & Smets, 2025) and enforcing equivariance on the vector fields (Kim et al., 2026; Wang et al., 2025). Recently, group symmetry-based diffusion and flow matching models have successfully tackled challenges across various domains, including molecular generation (Hoogeboom et al., 2022; Guan et al., 2023; Song et al., 2023a), and robotics (Ryu et al., 2024; Braun et al., 2024).

**Equivariant models for Novel View Synthesis.** Most of the previous works focus on enforcing equivariance at the representation or architectural level, often by explicitly encoding pose variables or constructing features that transform according to a predefined group action. These works do not directly address the 3D NVS considered in our work, where the model must infer unseen viewpoints and reason about self-occluded object regions from only partial observations. For example, Bekkers et al. (2024); Vadgama et al. (2022; 2023) address in-plane rotation problem, which does not require inferring about occluded views of a higher-dimensional object. Extending these to out-of-plane rotation would be not straightforward. Meanwhile, works such as Cohen & Welling (2016; 2017) develop architectures with built-in equivariance properties through group convolutions and steerable filters. In contrast, the NVS task studied in this paper requires synthesizing previously unseen observations from limited viewpoints, going beyond standard equivariant feature transformations.

**Generative Models for Novel View Synthesis.** Generative frameworks (Ho et al., 2020; Song et al., 2020; Goodfellow et al., 2020) have notably advanced NVS, prominently exemplified by NeRF-based models (Mildenhall et al., 2021; Lin et al., 2023; Yu et al., 2021). Recent models targeting NVS have achieved notable improvements in visual quality and view consistency (Karnewar et al., 2023; Yu et al., 2023; Shi et al., 2024; Anciukevičius et al., 2023; Ye et al., 2024; Chan et al., 2022). However, NeRF requires heavy integration of the continuous volumetric field. Our method offers a practical advantage in that it operates in a compact latent space

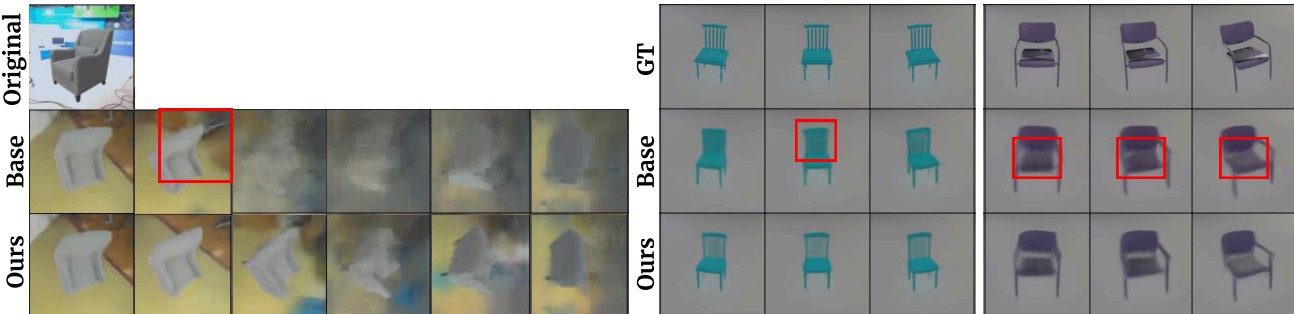

*Figure 6.* **Qualitative comparison on out-of-plane NVS** (SO(3)) **on OOD data.** *Left:* Results from the ABO-Material (OOD) with test-time SO(3) rotations without ground-truth. Base indicates NFT (Koyama et al., 2024). As the rotation angle grows, the baseline exhibits more corrupted renderings where the background is not well preserved. *Right:* Results from the ComplexBRDFs (OOD) with SO(2) rotations. Our method retains structural fidelity, capturing the fine details of the original image.

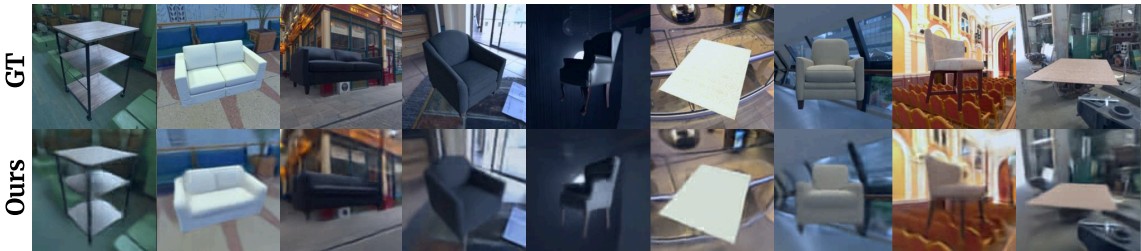

*Figure 7.* **Qualitative comparison on in-distribution NVS** (SO(3))**, ABO-Material.** Higher visual fidelity can be achieved for in-distribution viewpoints, on moderately high resolution images (224x224) with high-frequency details.

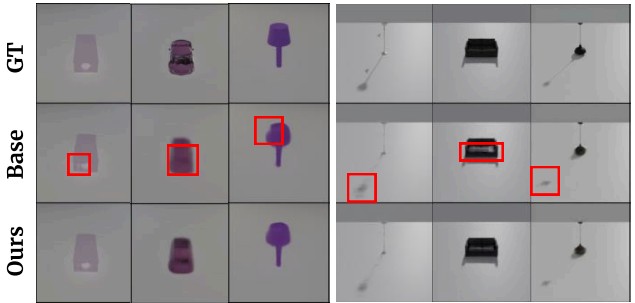

*Figure 8.* **Qualitative comparison on out-of-plane NVS** (SO(2)) **on OOD data.** *Left:* Results from the ComplexBRDFs OOD set. *Right:* Results from the ABO-Material Day-to-Night OOD set.

without requiring continuous field representations, dense multi-view supervision, or modifications to the rendering pipeline and this makes our approach more lightweight. In this sense, our method provides a general correction mechanism that enhances equivariance in realistic NVS pipelines.

## 6. Conclusion

We identify a practical limitation in equivariant representation learning: even when models are trained with explicit equivariance objectives, analytically transformed latents $\rho(g)\Phi(x)$ often fail to align with the true encoded targets $\Phi(g \circ x)$. We refer to this discrepancy as *latent misalignment*, which accumulates under transformations and de-

grades novel view synthesis quality. To address this, we propose Residual Latent Flow (RLF), a flow-based latent correction framework that learns a residual transport from analytically transformed latents to their corresponding target latents while preserving the underlying group-theoretic structure. Unlike conventional flow matching, our formulation explicitly models paired transport between latents corresponding to the same object under known group actions. Experiments on SO(n) demonstrate that our method consistently improves latent alignment, equivariance consistency, and reconstruction fidelity on both synthetic and real-image datasets, including out-of-distribution viewpoints and objects.

**Limitations.** Our method improves latent alignment, but the final image quality remains bounded by the decoder capacity itself. Incorporating stronger decoders, perceptual objectives, or multi-scale architectures could further improve reconstruction fidelity. In addition, our current framework focuses on controlled SO(2) and SO(3) transformations with known group actions. Extending the method to more general settings such as SE(3), articulated motion, or real-world unconstrained transformations remains an important future direction. Finally, developing stable end-to-end training strategies for jointly optimizing the encoder, flow correction module, and decoder may further enhance equivariant consistency and generation quality.

## Acknowledgments

This work was also supported by Samsung Electronics, Youlchon Foundation, National Research Foundation of Korea (NRF) grants (RS-2021-NR05515, RS-2024-00336576, RS-2023-0022663), and the Institute for Information & Communication Technology Planning & Evaluation (IITP) grants (RS-2022-II220264, RS-2024-00353131) funded by the Korean government.

## Software and Data

We provide complete details of our experimental setup, datasets, and model architectures and hyperparameters in Section 4 and Section F. Descriptions of both Residual Latent Flow correction framework and the procedures for generating synthetic rotational image datasets are given in the main text. Code is available at https://github.com/jaehoon-hahm/residual-latent-flow.

## Impact Statement

Our work focuses on developing techniques which are purely computational, relying exclusively on image datasets with controlled geometric variations. No human subjects, personal data, or sensitive contents are involved. We therefore identify no ethical concerns arising from this research. For this framework, there are many possible societal impacts, none of which need specific highlighting.

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

# Appendix

## A. Training Algorithm

---

**Algorithm 1** RLF Training

---

**Input:** Training sample distribution $p_{\text{data}}$, group $G$, Encoder $\Phi$, decoder $\Psi$, flow model $v_\theta$, stochasticity factor $\sigma$, number of integration steps $N$

1: **for** each training iteration **do**
2:      Sample $x \sim p_{\text{data}}$, $g \sim G$
3:      Minimize ERL loss to update encoder $\Phi$ and decoder $\Psi$:

$$\mathcal{L}_{\text{ERL}} = \|\Phi(g \circ x) - \rho(g)\Phi(x)\|_2^2 + \|g \circ x - \Psi(\rho(g)\Phi(x))\|_2^2$$

4: **end for**
5: Freeze encoder $\Phi$
6: **for** each training iteration **do**
7:      Sample $x \sim p_{\text{data}}$, $g \sim G$, $t \sim U[0,1]$, $\varepsilon \sim N(0,I)$
8:      Compute frozen source and target latents $z_0 = \rho(g)\texttt{sg}(\Phi)(x)$, $z_1 = \texttt{sg}(\Phi)(g \circ x)$
9:      Compute interpolation $z_t = (1-t)z_0 + tz_1 + \sigma^2\varepsilon$
10:     Minimize RLF loss $\mathcal{L}_{\text{RLF}}$ to update $\hat{\psi}_1$:

$$\mathcal{L}_{\text{RLF}} = \|v_\theta(z_t, t) - (z_1 - z_0)\|_2^2,$$

11: **end for**
12: Freeze flow model $v_\theta$
13: **for** each training iteration **do**
14:     Sample $x \sim p_{\text{data}}$, $g \sim G$
15:     Initialize source latent $z^{(0)} = \rho(g)\,\texttt{sg}(\Phi)(x)$
16:     **for** $n = 0, \ldots, N-1$ **do**
17:        $\tau_n = n/N$
18:        $z^{(n+1)} = z^{(n)} + \texttt{sg}(v_\theta)\big(z^{(n)}, \tau_n\big)/N$
19:     **end for**
20:     Compute corrected latent $\tilde{z} = z^{(N)}$
21:     Minimize fine-tuning loss to update decoder $\Psi$:

$$\mathcal{L}_{\text{fine-tune}} = \|g \circ x - \Psi(\tilde{z})\|_2^2$$

22: **end for**

---

## B. Classes of Novel View Synthesis

Novel View Synthesis (NVS) (Miyato et al., 2022; Koyama et al., 2024; Miyato et al., 2024) is a task to generate realistic images of a specific subject or scene from a specific point of view, given a set of images for the same scene taken from different viewpoints. The NVS task can be categorized into different classes by the types of allowed rotation of the object (or equivalently, the camera viewpoint). Figure I illustrates several representative classes of the NVS tasks. The in-plane rotation shown in (a) allows the object (or the camera) to rotate around an axis that is parallel to its viewing direction. Since this is equivalent to rotating the resulting images in the same 2D space, this is a relatively easy task. Out-of-plane rotations, on the other hand, occur when the camera view and rotation axis are not aligned. In particular, out-of-plane rotation under SO(2) freedom shown in (b) allows the rotation axis not to be in parallel to the viewing direction. Thus, the area and shape of the object in the projected 2D image varies by the rotation angle, making it significantly more challenging compared to the in-plane rotation. The most challenging scenario is shown in (c), where the camera can move in the 3D space both latitudinally and longitudinally at the same time under rotational freedom of SO(3), requiring the model to predict views from arbitrary angles. In this paper, we aim to tackle all of these challenging NVS task, not just the simple in-plane rotation problem.

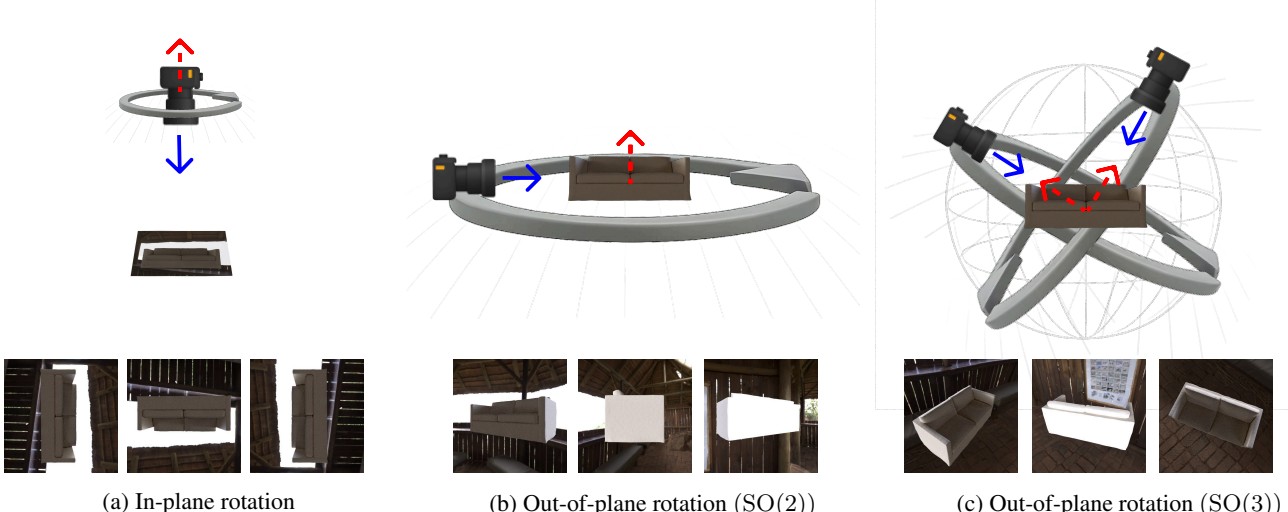

(a) In-plane rotation      (b) Out-of-plane rotation $(SO(2))$      (c) Out-of-plane rotation $(SO(3))$

*Figure 1.* **Illustration depicting different classes of Novel View Synthesis (NVS) tasks.** Blue arrow indicates camera's viewing direction and red arrow indicates the axis of rotation. (a) In-plane rotation: the camera rotates around an axis parallel to the viewing direction. (b) Out-of-plane rotation with $SO(2)$ freedom: the camera rotates around an axis that is not parallel to the viewing direction. (c) Out-of-plane rotation with full $SO(3)$ freedom: the camera can move in an arbitrary direction in 3D space, representing the most challenging NVS scenario.

## C. Wigner $D$-matrix representation

**Conventions and indices.** We use the active Z–Y–Z Euler convention for $g \in SO(3)$ with angles $(\alpha, \beta, \gamma)$. Fix an integer degree $\ell \in \{0, 1, 2, \dots\}$ and the index set $\mathcal{I}_\ell = \{-\ell, \dots, \ell\}$. All $(2\ell+1) \times (2\ell+1)$ matrices below are written in the spherical basis ordered by rows/columns $m, n \in \mathcal{I}_\ell$. The generators of $SO(3)$, $J_x^{(\ell)}, J_y^{(\ell)}, J_z^{(\ell)}$, (Hermitian matrices acting on this space) satisfy $[J_x, J_y] = iJ_z$ and cyclic permutations.

**Definition.** The Wigner $D$-matrix is the matrix exponential product
$$D^{(\ell)}(g) = \exp(-i\alpha J_z)\exp(-i\beta J_y)\exp(-i\gamma J_z) \in \mathbb{C}^{(2\ell+1)\times(2\ell+1)},$$
with entries $D_{m,n}^{(\ell)}(g) = [D^{(\ell)}(g)]_{mn}$. In Z–Y–Z convention, $D^{(\ell)}$ factorizes elementwise as
$$D_{m,n}^{(\ell)}(\alpha, \beta, \gamma) = e^{-im\alpha}\, d_{m,n}^{(\ell)}(\beta)\, e^{-in\gamma},$$
and the *small-d* matrix $d^{(\ell)}(\beta) = \exp(-i\beta J_y)$ is real.

**Closed form for the small-$d$ entries.** With row/column order $(m, n)$,
$$d_{m,n}^{(\ell)}(\beta) = \sqrt{(\ell+m)!\,(\ell-m)!\,(\ell+n)!\,(\ell-n)!}\ \sum_{s=s_{\min}}^{s_{\max}} \frac{(-1)^{m-n+s}\,(\cos\frac{\beta}{2})^{2\ell+n-m-2s}\,(\sin\frac{\beta}{2})^{m-n+2s}}{(\ell+n-s)!\,s!\,(m-n+s)!\,(\ell-m-s)!},$$
where $s_{\min} = \max(0, n-m)$ and $s_{\max} = \min(\ell+n, \ell-m)$.

**Real basis and $\ell{=}1$ block.** A fixed change of basis $Q_\ell$ gives $D_{\mathrm{real}}^{(\ell)}(g) = Q_\ell D^{(\ell)}(g) Q_\ell^\top$ (real orthogonal). For $\ell = 1$, $D_{\mathrm{real}}^{(1)}(g)$ equals the standard $3 \times 3$ rotation $R(g)$.

If we restrict to SO(2) rotations (i.e. $\beta = \gamma = 0$ and $\alpha = \theta$), the Wigner $D$-matrices reduces to
$$D_{m,n}^{\ell}(\theta, 0, 0) = \delta_{m,n}\, e^{-im\theta}. \tag{12}$$

Thus the $(2\ell + 1)$-dimensional irrep of SO(3) decomposes under restriction as
$$D^\ell(\theta)\Big|_{\mathrm{SO}(2)} \cong \rho_{-\ell}(\theta) \oplus \rho_{-\ell+1}(\theta) \oplus \cdots \oplus \rho_\ell(\theta). \tag{13}$$

In a real basis, the complex conjugate pair $\rho_m$ and $\rho_{-m}$ can be combined into a $2 \times 2$ rotation block:

$$R_m(\theta) = \begin{pmatrix} \cos(m\theta) & -\sin(m\theta) \\ \sin(m\theta) & \cos(m\theta) \end{pmatrix}, \qquad m \geq 1, \tag{14}$$

together with the trivial representation $R_0(\theta) = [1]$. We will use the direct product of these real representations $\phi(g) = \bigoplus_{m=0}^{n-1} \phi_m(g)$ as the group representation of $\mathrm{SO}(2)$.

Our representation decomposes as

$$B(g) = P\,\rho(g)\,P^{-1} = \bigoplus_{\ell=0}^{L} \left( D^{(\ell)}(g) \otimes I_{m_\ell} \right),$$

so the $\ell$-th latent block transforms exactly by $D^{(\ell)}(g)$ (or its real form); we exploit the $\ell = 1$ block for the Wahba alignment in the main text.

## D. Wahba Problem

Given weighted direction correspondences $\{(r_i, b_i, a_i)\}_{i=1}^n$ with $a_i > 0$ and $\sum_i a_i = 1$, Wahba's problem (Markley, 1988) seeks the proper special orthogonal matrix (rotation matrix) $A \in \mathrm{SO}(3)$ minimizing

$$L(A) = \tfrac{1}{2} \sum_{i=1}^n a_i \|b_i - Ar_i\|^2 \;=\; 1 - \mathrm{tr}(AB^\top), \qquad B = \sum_{i=1}^n a_i\, b_i r_i^\top. \tag{15}$$

Let the singular value decomposition be $B = U\,S\,V^\top$ with $S = \mathrm{diag}(s_1, s_2, s_3)$, $s_1 \geq s_2 \geq s_3 \geq 0$. Define $d = \det(U)\det(V) \in \{+1, -1\}$. Then Markley's SVD solution gives

$$A_{\mathrm{opt}} \;=\; U\,\mathrm{diag}(1, 1, d)\,V^\top \;\in\; \mathrm{SO}(3), \tag{16}$$

which minimizes $L(A)$. Intuitively, writing $W = U^\top A V$ yields $L(A) = 1 - \mathrm{tr}(S'W)$ with $S' = \mathrm{diag}(s_1, s_2, d\,s_3)$, and the minimum occurs at $W = I$.

**Properties and uniqueness.** If $\mathrm{rank}(B) \geq 2$ (i.e., $s_2 > 0$), the solution is unique except in the degenerate limit where $B$ is near rank $< 2$; in that case a one-parameter family of minimizers appears (rotation about an axis). The SVD approach is numerically robust (avoids squaring $B$) and, unlike certain fast implementations of Davenport's $q$-method, naturally exposes eigen-structure used for covariance analysis; see Markley (1988) for closed-form covariance expressions.

This is a procedure for solving the Wahba problem:

1. Form $B = \sum_i a_i b_i r_i^\top$ (normalize $\sum_i a_i = 1$).

2. Compute $B = USV^\top$

3. Set $d = \det(U)\det(V)$.

4. Return $A_{\mathrm{opt}} = U\,\mathrm{diag}(1, 1, d)\,V^\top$ (guarantees $\det(A_{\mathrm{opt}}) = +1$).

## E. Equivariance Error Metrics for $\mathrm{SO}(2)$ and $\mathrm{SO}(3)$

### E.1. Equivariance Error for $\mathrm{SO}(2)$

For $\mathrm{SO}(2)$, we estimate the relative rotation angle between $\Phi(g_\theta \circ x)$ and $\rho(g_\theta)\Phi(x)$ in a degree-wise manner. For each degree-$\ell$ block, we solve

$$\delta\hat{\theta}_\ell = \arg\min_{\delta\theta} \|\Phi_\ell(g_\theta \circ x) - R_\ell(\delta\theta)\,\Phi_\ell(x)\|_F^2, \tag{17}$$

where $R_\ell(\theta)$ denotes the degree-$\ell$ block of $\rho(g_\theta)$ and $\Phi_\ell(\cdot)$ denotes the degree-$\ell$ representation.

Higher-degree $SO(2)$ representations admit degenerate angle solutions. To obtain a stable estimate, we select the solution with the smallest magnitude for each $\hat{\delta\theta}_\ell$. We then aggregate the estimates across degrees,

$$\hat{\delta\theta} = \frac{1}{L}\sum_{\ell=1}^{L}\hat{\delta\theta}_\ell. \tag{18}$$

The equivariance error is measured using an angular cosine distance,

$$d_{\cos}(\hat{\delta\theta}) = 1 - \cos(\hat{\delta\theta}). \tag{19}$$

### E.2. Equivariance Error for $SO(3)$

For $SO(3)$, only the degree-1 representation is invertible with respect to the underlying rotation parameters, whereas higher-degree Wigner $D$ blocks provide equivariant embeddings that do not uniquely determine $(\alpha, \beta, \gamma)$. We therefore estimate the relative rotation using the degree-1 block by solving a Wahba problem,

$$\hat{R}(\alpha, \beta, \gamma) = \underset{R \in SO(3)}{\arg\min} \left\| \Phi_{\ell=1}(g_{\alpha,\beta,\gamma} \circ x) - R(\alpha, \beta, \gamma)\, \Phi_{\ell=1}(x) \right\|_F^2, \tag{20}$$

where $R(g)$ denotes the degree-1 block of $\rho(g)$. Details of the numerical solver are provided in Appendix Section D.

For perfect equivariance, the relative rotation between $\Phi(g \circ x)$ and $\rho(g)\Phi(x)$ should be the identity. We convert the estimated rotation $\hat{R}$ into a quaternion $\hat{q}$ and measure its deviation from the identity quaternion $q_{\text{id}} = (1, 0, 0, 0)$ using

$$d_{\cos}(\hat{q}, q_{\text{id}}) = 1 - |\langle \hat{q}, q_{\text{id}} \rangle|. \tag{21}$$

For higher-degree $SO(3)$ representations, which are not invertible with respect to the rotation parameters, we instead report a latent error computed as the $\ell_2$ distance between the predicted latents and the corresponding ground-truth latents.

## F. Implementation Details

### F.1. Datasets

**ABO-Material.** This dataset (Collins et al., 2022) is based on rendered images from the Amazon-Berkeley Objects (ABO) collection. It contains 7,678 distinct objects, each rendered from 91 viewpoints uniformly distributed over the upper hemisphere of an icosphere, covering variations in both azimuth and elevation. Each object is rendered under three different high-dynamic-range (HDR) environment maps with varying lighting conditions and backgrounds, resulting in a total of $7{,}678 \times 3 \times 91$ images.

**ModelNet10-SO(3).** This dataset (Liao et al., 2019) contains clean, object-centric renderings of CAD models from the ModelNet10-SO(3) benchmark. Objects are viewed from random $SO(3)$ rotations without background or environmental textures, providing a minimal setting for analyzing pure rotational equivariance.

The training set consists of 3,991 objects, each rendered from 100 randomly sampled viewpoints, resulting in a total of 399,100 training images. For out-of-distribution (OOD) evaluation, we use 908 unseen objects, each rendered from 20 viewpoints, yielding 18,160 OOD images.

**ComplexBRDFs.** This dataset (Greff et al., 2022) comprises object-only renderings of ShapeNet models under complex materials (e.g., metallic, glossy). It contains 49,198 objects. Each object undergoes a full 360° in-plane rotation (about the z-axis), sampled at 24 evenly spaced steps, resulting in a total of $49{,}198 \times 24$ images. This yields a structured $SO(2)$ transformation setting, focusing on view consistency under material-induced appearance variation.

**ABO-Material Day-to-Night.** To evaluate generalization beyond rigid geometric transformations, we introduce a variant of the ABO-Material dataset (Collins et al., 2022). Each object is rendered from a fixed viewpoint while lighting direction changes along a 170° arc in 10° increments, resulting in a total of $7{,}678 \times 18$ images. This results in 18 lighting conditions per object, simulating a structured day-to-night transition with shadows and specular variation. This defines a quasi-$SO(2)$ transformation in appearance space, without explicit rotation of object geometry.

**RotatedMNIST.** RotatedMNIST is a planar rotation variant of the MNIST dataset (Deng, 2012), consisting of 70,000 handwritten digit images. Each digit is rotated in-plane at fixed angular intervals of $15°$, yielding 24 rotated views per original image that uniformly cover the full $SO(2)$ rotation range.

**SmallNORB.** The SmallNORB dataset (LeCun et al., 2004) contains images of 50 physical toy objects captured using a real camera under controlled conditions. Each object instance is photographed across 18 azimuth angles, 9 elevations, and 6 lighting directions, resulting in a total of 48,600 images. It produces systematic variations in viewpoint and illumination. As the images stem from actual camera captures rather than synthetic renderings, the dataset provides a realistic benchmark for testing equivariance and robustness to real-world visual factors.

We follow an object-level split, where 50% of the object instances are held out as an out-of-distribution (OOD) set, and the remaining objects are used for training.

### F.2. Dataset Splits.

For datasets without predefined splits, including ABO-Material (Day-to-Night), RotatedMNIST and ComplexBRDFs, we construct the evaluation protocol in a unified manner. We first reserve 5% of object instances as an out-of-distribution (OOD) set. From the remaining images, 10% are used as a test set, and the rest are used for training. For datasets with predefined splits, such as ModelNet10-SO(3) and SmallNORB, we follow the official dataset splits without modification.

### F.3. Shape of latent representations

The latent representation shapes depend on the underlying symmetry group: for $SO(3)$, the latents have a shape of $C \times 81$, corresponding to the block-diagonal Wigner-$D$ representation with degrees $\ell = 0, \ldots, 8$. We use $C = 128$ for ABO-Material and SmallNORB, and $C = 64$ for ModelNet10-SO(3). For processing with a U-Net, the latents are reshaped to $C \times 9 \times 9$. For $SO(2)$, these latents have a shape of $C \times 17$. We use $C = 128$ for ComplexBRDFs and ABO-Material Day-to-Night, and $C = 64$ for RotatedMNIST.

## G. Transfer to Rotation Estimation

*Table I.* **Rotation estimation on ABO Day-to-Night.**

| Method | Test Error | OOD Error |
|---|---|---|
| NFT | 0.002507 | 0.004473 |
| **Ours** | **0.001131** | **0.001961** |

To test whether the latent representations that better preserve the underlying rotational structure improves the performance in other downstream tasks such as pose estimation, we compare the performance on the rotation angle prediction task using ABO-Material Day-to-Night.

Given a latent feature tensor $z \in \mathbb{R}^{64 \times 17}$ produced by the encoder, we attach a light regression head consisting of two fully-connected layers with ReLU activations, followed by a linear output layer that predicts a rotation angle $\theta_0$ of the given image. We only train the regression head while keeping the latent encoder frozen.

For the NFT baseline, the latent rotation is obtained analytically through the group action $D(\Delta\theta)z$, and the regressor is trained to recover the target angle $\theta_1 = \theta_0 + \Delta\theta$. For our method, the analytically rotated latent is further refined through the learned flow module before angle prediction. We measure accuracy using the cosine-based angular discrepancy $1 - \cos(\hat{\theta}_1 - \theta_1)$.

Table I compares the rotation angle estimation performance of our method and that of the NFT baseline. We observe that our method achieves significantly lower angular discrepancy on both the test and OOD sets, verifying that our approach is applicable to this different downstream task.

## H. Ablation on Noise Level of Stochastic Path

*Table II.* **Stochastic Interpolation Results with Varying Noise Levels.**

| Stochasticity ($\sigma$) | Latent Error | PSNR | Prediction Error |
|---|---|---|---|
| **0** | **0.0002607** | **22.84** | **0.0065** |
| 0.01 | 0.0002872 | 22.68 | 0.0067 |
| 0.05 | 0.0003631 | 22.22 | 0.0074 |
| 0.1 | 0.0005406 | 19.30 | 0.0129 |

To construct a stochastic interpolant (Albergo et al., 2025), we add a stochastic correction to the velocity instead of injecting noise into the state. Let $x_0$ and $x_1$ be the endpoints. The deterministic linear interpolant is $\mu_t = (1-t)x_0 + tx_1$, representing the mean trajectory, with velocity $v_{\text{det}} = x_1 - x_0$.

The stochastic correction, derived from the Brownian bridge drift, keeps the trajectory anchored at the endpoints:

$$v_{\text{sto}}(t) = \frac{1-2t}{2t(1-t)}(x_t - \mu_t).$$

The full velocity is $v(t) = v_{\text{det}} + \sigma \, v_{\text{sto}}(t)$, where $\sigma$ controls stochastic strength. Setting $\sigma = 0$ recovers the deterministic interpolant, while larger $\sigma$ increases trajectory variability.

In our ablation study, we vary $\sigma \in \{0, 0.01, 0.05, 0.1\}$ and measure latent reconstruction error, PSNR, and prediction error. Our ablation results over the noise scale in Table II indicate that a larger stochasticity gradually degrades latent reconstruction, PSNR, and prediction accuracy. In other words, since our objective is to learn a precise deterministic correction path rather than to model a high–entropy family of trajectories, strong stochastic perturbations of the interpolant are not beneficial in this regime and can even hinder optimization.

Taking a deeper look, under our setting, the flow is used purely as a deterministic correction map from a misaligned latent $z_0$ to its aligned counterpart $z_1$. Thus, the underlying conditional distribution $p(z_1 \mid z_0)$ is effectively low–entropy and close to a one-to-one mapping. While the stochastic interpolant framework of Albergo et al. (2025) allows one to introduce nontrivial diffusion along the path without changing the endpoint marginals, this additional stochasticity does not enrich the target distribution in our correction scenario. Instead, increasing the noise level merely enlarges the variance of the training trajectories $x_t$ around the same endpoints, which acts as label noise for the velocity field.

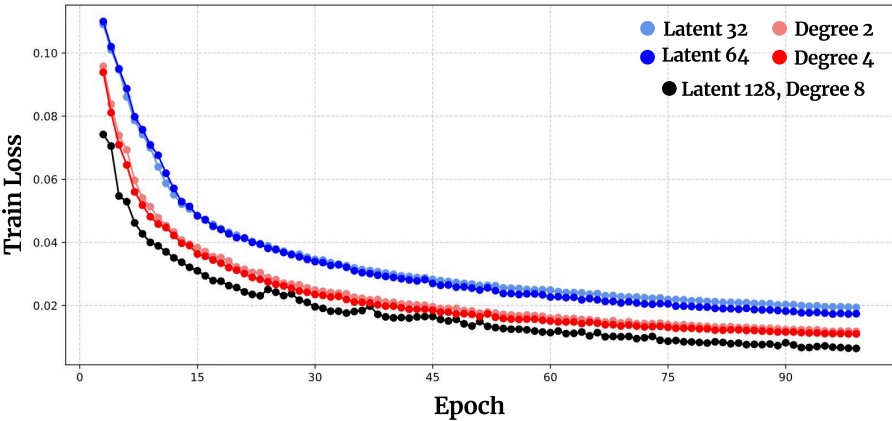

*Figure II.* **Experiment on Scaling.** The NFT's train loss converges stably across different latent scales, demonstrating that the NFT baseline remains reliable under varying dimensional settings.

## I. Experiment on Scalability

*Table III.* **Performance as latent channel dimension $C$ varies with fixed max degree $L = 8$.**

| $C$ | Method | PSNR | Latent Error | Pred Error | Latency (ms/sample) | Params (M) |
|---|---|---|---|---|---|---|
| 32 | Base | 20.20 | 0.00008 | 0.0100 | 0.969 | 10 |
| | Ours | 20.56 | 0.00006 | 0.0092 | 1.265 | 10 + 2 |
| 64 | Base | 20.79 | 0.00013 | 0.0087 | 0.979 | 32 |
| | Ours | 21.12 | 0.00010 | 0.0081 | 1.310 | 32 + 2 |
| 128 | Base | 22.73 | 0.00019 | 0.0056 | 1.298 | 120 |
| | Ours | 24.32 | 0.00015 | 0.0039 | 1.785 | 120 + 2 |

*Table IV.* **Performance as maximum representation degree $L$ varies with fixed latent channel $C = 128$.**

| $L$ | Method | PSNR | Latent Error | Pred Error | Latency (ms/sample) | Params (M) |
|---|---|---|---|---|---|---|
| 2 | Base | 22.34 | 0.00009 | 0.0061 | 1.155 | 101 |
| | Ours | 22.73 | 0.00008 | 0.0056 | 1.743 | 101 + 2 |
| 4 | Base | 22.48 | 0.00008 | 0.0060 | 1.179 | 107 |
| | Ours | 22.99 | 0.00007 | 0.0053 | 1.758 | 107 + 2 |
| 8 | Base | 22.73 | 0.00019 | 0.0056 | 1.298 | 120 |
| | Ours | 24.32 | 0.00015 | 0.0039 | 1.785 | 120 + 2 |

**Inference Time Measurement.** All models are benchmarked on a single NVIDIA RTX A6000 GPU in `eval` mode with all parameters frozen. For each configuration, we perform one warm-up validation pass and then measure the runtime of a second `validate()` call, placing `torch.cuda.synchronize()` before and after the measurement for accurate GPU timing. The per-sample latency is obtained by dividing the total validation time by the number of samples in the OOD split. For flow models, latent correction was performed using a fixed 10-step generation procedure.

**Flow Matching Training Cost.** To quantify the computational overhead of our flow module, we additionally summarize its training-time characteristics. On the SO(2) RotMNIST setting, the flow model runs at approximately 65ms per optimization step with batch size 256 and uses about 1.96GB of GPU memory on a single NVIDIA RTX A6000. On the SO(3) ABO setting, the corresponding flow model takes roughly 110ms per step with the same batch size, and consumes around 2.15GB of GPU memory.

*Table V.* **Effect of noisy rotation labels on the ABO-Material Day-to-Night dataset.**

| Noise level | Split | NFT (PSNR) | Ours (PSNR) |
|---|---|---|---|
| $10.0°$ | OOD | 17.39 | 17.52 |
| $10.0°$ | Test | 17.82 | 17.95 |
| $7.5°$ | OOD | 18.81 | 19.08 |
| $7.5°$ | Test | 19.27 | 19.56 |
| $5.0°$ | OOD | 20.60 | 20.90 |
| $5.0°$ | Test | 21.17 | 21.53 |
| $2.5°$ | OOD | 21.40 | 21.83 |
| $2.5°$ | Test | 21.85 | 22.56 |
| $0°$ | OOD | 21.80 | 22.84 |
| $0°$ | Test | 22.73 | 24.32 |

## J. Experiment under Noisy Label Setting

We corrupt the labels by up to $k°$, with $k \in \{2.5, 5.0, 7.5, 10.0\}$ on the 10% of the training samples. For each noise level, we report PSNR on the OOD and test splits in Table V, comparing with the NTF baseline with our flow-corrected model. We observe that our method consistently outperforms the NFT baseline both on the OOD and test split across all the tried noise levels. Also, the performance degradation is mild enough to use in practice, unless the noise level is relatively high (*e.g.*, larger than $10°$).

## K. Why is there Latent Misalignment? - Conflicting Loss Objectives in ERL

The equivariance loss enforces the latent of a transformed image $\Phi(g \circ x)$ to match the analytically rotated latent $\rho(g)\Phi(x)$, thereby promoting equivariance of the encoder. At the same time, the decoder loss optimizes autoencoder for precise reconstruction by encouraging the decoder to decode latent $\Psi(\rho(g)\Phi(x))$ to match the ground truth of the transformed image $g \circ x$.

Correcting this latent misalignment is crucial, since it undermines the consistency and interpretability of the learned representations. The latent misalignment naturally arises as the analytical transformation $\rho(g)$ is a fixed linear operator applied in the latent space, it lacks the expressiveness to model fine-grained visual effects such as self-occlusion, lighting variation, or subtle texture changes that arise from real 3D transformations.

Ideally, if the encoder were perfectly trained, the encoded latent would behave well-aligned with the group actions, where latent trajectories along the angle (latent trajectory) under linear transformations align cleanly along a fixed orbit $\text{Orb}(x) := \{g \circ x \mid g \in G\}$ generated by the defined group actions. As illustrated in the Figure 3, the degree-1 representations on $SO(3)$ would ideally trace smooth circular path confined to the surface of the sphere, such that trajectories originating from different starting viewpoints converge to the same target view. In practice, however, the paths deviate from the ideal spherical orbit. The latents scatter off the surface, and applying transformation from different viewpoints no longer leads to a consistent target.

While the encoder should preserve the structure of group transformations via $\mathcal{L}_{\text{equiv}}$, it also needs to encode sufficient object detail (*e.g.*, texture, shape) to enable faithful image reconstruction to minimize $\mathcal{L}_{\text{decoder}}$. For example, consider a set of images depicting a perfectly uniform sphere with uniform lighting. Any rotation about the $z$-axis results in visually indistinguishable images. From a reconstruction standpoint, the encoder would assign an identical latent representation for every view. Meanwhile, to satisfy the equivariance relation, the model must encode every viewpoints differently, as they correspond to distinct group elements. This leads to conflicting gradients, resulting in the imperfectness of the latent encoding in ERL.

# L. Additional Comparisons

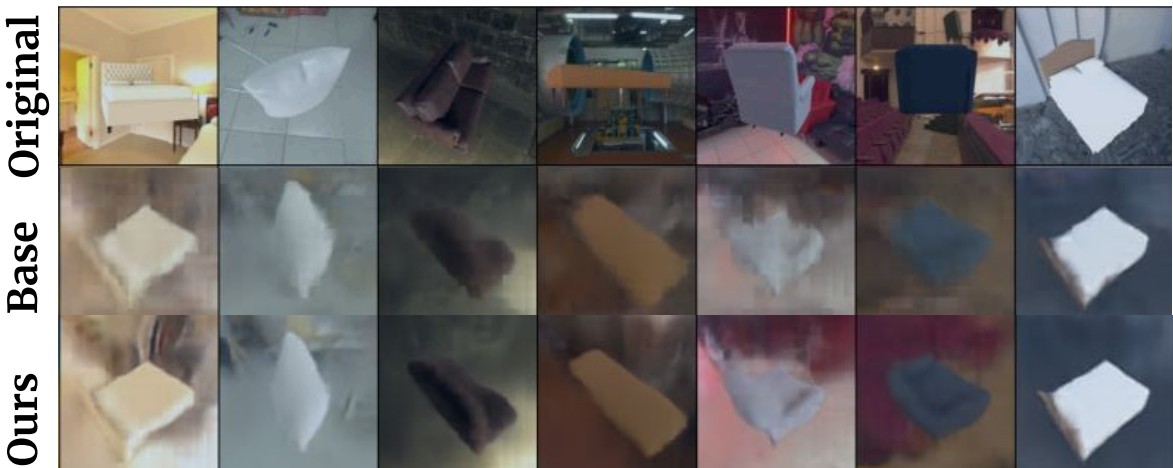

*Figure III.* Qualitative comparison of novel view synthesis on out-of-distribution (OOD) datasets with and without latent correction. Results from the ABO-Material OOD set.

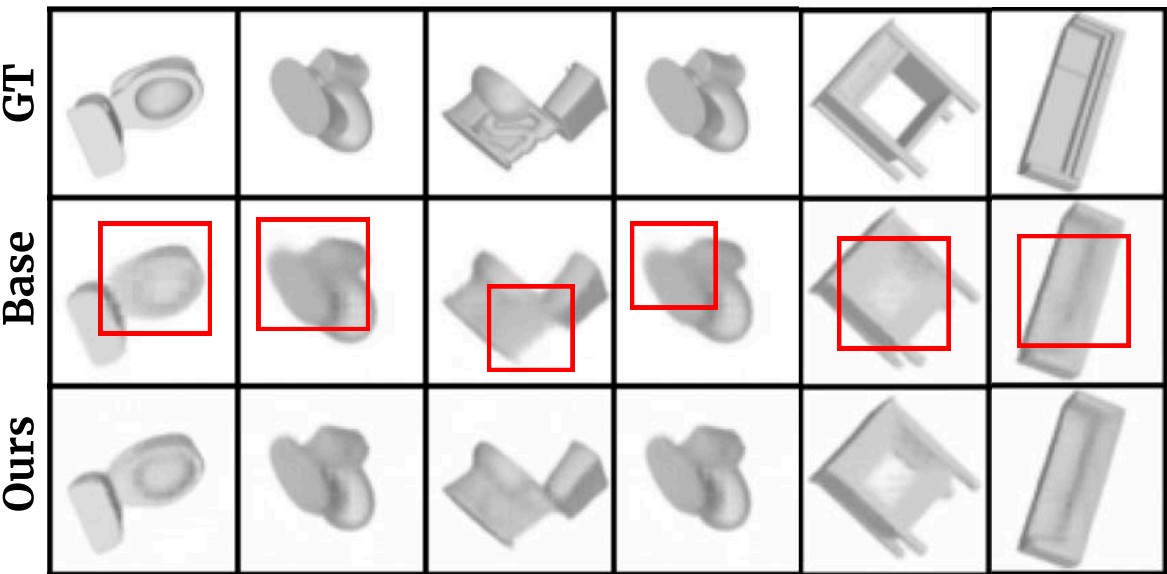

*Figure IV.* Qualitative comparison of novel view synthesis on out-of-distribution (OOD) datasets with and without latent correction. Results from the ModelNet10-SO(3) OOD set.

