# OpenReview forum: "Equivariant Latent Alignment via Flow Matching under Group Symmetries"
_ICML.cc/2026/Conference — ICML 2026 regular_

### Official Review · Reviewer_ddXH · 2026-02-26

**Soundness:** 3
**Presentation:** 3
**Significance:** 2
**Originality:** 3
**Overall Recommendation:** 4
**Confidence:** 2

**Summary:**

This paper introduces Residual Latent Flow, a flow-based architecture designed to rectify misaligned latent representations and enforce stricter adherence to the underlying equivariant constraints.

**Compliance With Llm Reviewing Policy:**

Affirmed.

**Final Justification:**

My concerns have been basically addressed.

**Key Questions For Authors:**

1. Does the proposed method offer distinct advantages compared to current diffusion-based approaches?
2. What are the potential challenges in scaling this method to high-resolution outputs or large-scale scenes?

**Limitations:**

yes

**Strengths And Weaknesses:**

# Strengths:

1. The paper provides rigorous mathematical proofs and theoretical grounding.

2. Experimental results across multiple datasets demonstrate substantial improvements over the NFT baseline.

3. The qualitative visualizations further validate the effectiveness of the proposed method.

# Weaknesses:

1. The baselines compared are somewhat outdated; NFT is a 2023 method, and the paper lacks discussion of more recent literature.

2. The visualization results suffer from low resolution, and the target objects are relatively simple. This lags behind current diffusion-based methods and remains a distance away from practical engineering utility.

---

> ### Author Rebuttal · Authors · 2026-03-31
>
> We appreciate the reviewer for their thoughtful feedback.
>
> ---
>
> __[W1. Additional baselines, generalization beyond NFT]__
>
> To demonstrate that our correction method generalizes across various equivariant representation spaces, we further evaluate it using LGA [A] and observe a substantially reduced latent mismatch. Specifically, the latent error decreases from 10.58 to 0.15 on RotMNIST, and from 1258 to 273 on SmallNORB. Since latent error directly measures the inconsistency between predicted rotated latent and the latent of ground-truth transformed image, these results provide the evidence that our method also effectively corrects the misaligned latents in LGA [A].
>
> [A] Jin, Y., Shrivastava, A., and Fletcher, P. T. Learning group actions on latent representations. Advances in Neural Information Processing Systems, 37:127273–127295, 2024.
>
> ---
>
> __[Q1. Comparison with diffusion-based approaches]__
>
> Our method can be viewed as a single-step NVS approach in latent space (excluding the encode–decode stage), in contrast to diffusion-based methods such as [A], which require iterative denoising (tens to hundreds of steps) combined with 3D-aware rendering or ray integration. As a result, our approach is inherently more efficient at inference time, as it avoids both diffusion sampling loops and heavy volumetric rendering procedures.
>
> Moreover, diffusion-based NVS methods rely on a pretrained score network trained on large-scale datasets (e.g., Stable Diffusion-scale models) to infer plausible 3D structure from limited views. This introduces substantial memory overhead, and ties performance to the availability of large pretrained models. In contrast, our method is lightweight and model agnostic; it operates with a light autoencoder backbone and leverages analytic group actions with a learned residual correction, requiring significantly less pretraining, and can be applied to any encoder-decoder frameworks for NVS.
>
> While we do not provide a full wall-clock comparison due to the time limit during rebuttal, we emphasize that our method scales with a few or even a single forward pass in latent space, whereas diffusion NeRF-based approaches typically require $\mathcal{O}(T)$ network evaluations (with $T \approx 50\text{–}100$) per sample, in addition to heavy rendering costs. We will clarify this computational distinction and include empirical comparisons in the final version.
>
> [A] Generative Novel View Synthesis with 3D-Aware Diffusion Models
>
> ---
>
> __[Q2. Potential challenges in scaling to high-resolution experiments]__
>
> We agree that the current visualizations are limited in resolution and object complexity, and we want to emphasize that this limitation primarily stems from the underlying autoencoder and latent representation rather than the proposed Residual Latent Flow (RLF) itself. Since RLF operates entirely in latent space, its performance is fundamentally bounded by what the encoder $\Phi$ can represent. In practice, standard autoencoders compress high-dimensional images into compact latents, which inevitably discard high-frequency details necessary for high-resolution synthesis. Moreover, these encoders often entangle geometric transformations with appearance, making it increasingly difficult to enforce consistent equivariance as scene complexity grows, which is the exact problem we are trying to resolve in this paper.
>
> Importantly, this does not contradict the success of diffusion-based methods, such as [A], which levarage pretrained Stable Diffusion, achieve high resolution through hierarchical latent spaces and powerful decoders. Our method about rectifying the latents via flow model is orthogonal to such advances and is fully compatible with stronger latent backbones, such as Latent Diffusion Models. We expect that adopting higher-capacity or multi-scale latent representations would directly alleviate the current limitations without requiring changes to the RLF framework itself. We will discuss more about this in the revised manuscript.

---

> > ### Author Rebuttal · Reviewer_ddXH · 2026-04-05
> >
> > Dear authors,
> > Thank you for your detailed and thoughtful response. You mentioned that the proposed method is fully compatible with existing diffusion-based approaches. Does this mean that combining RLF with diffusion models could deliver even greater advantages? Unfortunately, this potential is not demonstrated in the current paper.
> > Furthermore, the proposed RLF has only been validated on low-resolution, simple scenes, which unfortunately lacks sufficient persuasiveness.
> >
> > -------
> >
> > Thanks for the detailed response. I think my concerns are basically addressed, so I have decided to raise my rating.

---

> > > ### Author Response · Authors · 2026-04-08
> > >
> > > Thank you for the follow-up question!
> > >
> > > ---
> > >
> > > __[R1. Clarification on compatibility with diffusion]__
> > >
> > > First, our statement regarding compatibility with diffusion-based approaches was not intended to suggest that RLF could be directly combined with diffusion-based methods. In fact, diffusion-based approaches have no need to acquire an equivariant representation since they do not explicitly utilize the group structure. Our intent was to highlight that RLF can leverage well-established latent encoder architectures commonly adopted in such methods, e.g., SD-VAE, and our method can be generally combined with ERL frameworks to obtain better aligned latents.
> > >
> > > The main bottleneck of the final output quality is how well the encoder-decoder is optimized, and this is an orthogonal component with the proposed RLF method. By extensively training on a large image dataset as in Stable Diffusion, we presume the method could leverage the high-quality latents, and RLF can in principle capture fine-grained details and extend to higher-resolution outputs.
> > >
> > > ---
> > >
> > > __[R2. Advantage of ERL]__
> > >
> > > Second, we emphasize that our method is fundamentally different from diffusion-based approaches. Diffusion models aim to sample a plausible novel view that is consistent with a given image by following a learned score function, whereas we first learn an equivariant representation that respects group structure and then apply group-theoretic transformations directly in latent space, treating NVS as a downstream task.
> > >
> > > This distinction provides two key advantages. First, equivariance improves generalization. In testing on unseen-objects (OOD), learning a representation of a sample $x$ implicitly enforces consistency with its transformed counterpart $gx$. This shared structure reduces sample complexity and leads to better generalization across viewpoints, as supported by Tab. 1 and Fig. 6, 7. Second, our method enables deterministic and controllable transformations by directly applying group actions in latent space, which enables light inference, rather than relying on iterative sampling using score function in the latent space.
> > >
> > > ---
> > >
> > > __[R3. On dataset scale and resolution]__
> > >
> > > Regarding the concern about validation on relatively simple scenes, we acknowledge that our experiments are conducted on moderate-scale datasets. This is consistent with prior work in equivariant representation learning [1, 2, 3, 4, 5], where the primary objective is to study group structure and equivariance, rather than large-scale photorealistic synthesis. In fact, these existing works have relied on even simpler datasets such as Rotated MNIST, ShapeNet, Shapes3D, low-resolution cell/galaxy images, simple molecular conformation, and toy shapes with permutation. Provided these previous works focus on these dataset, we believe that the presented experimental results are dealing with a data that has sufficiently high-resolution and fine details (e.g. ABO), under ERL framework.
> > >
> > > Extending to larger-scale datasets introduces additional factors to consider beyond the core contribution. For example, CO3D is a human-filmed video revolving around the objects, which involves translation and rotation with inconsistent speed. This increases complexity compared to the well-controlled datasets adopted in our paper. We therefore focus on programmatically simulated datasets to maintain a controlled environment and isolate the effect of equivariant representation learning. Similarly, Matterport3D operates at a comparable hardness and effective resolution to ABO. Although we believe our method would eventually yield similar behavior on these larger datasets, we respectfully argue that handling those practical concerns requires substantial engineering burdens irrelevant to our core claim rather than providing new scientific discoveries or insights, considering this short rebuttal timeframe.
> > >
> > > Additionally, we emphasize that Fig. 6, 7 demonstrates generalization to unseen view angles, where test viewpoints are sampled outside the training range, which was focused as the primary objective of the work. Higher visual fidelity is achieved for in-distribution viewpoints. For support, we include additional results on ABO-material in this anonymous link, which features high-frequency textures and moderately high image resolution (224×224), to support the robustness and generation quality of our approach: https://tinyurl.com/3wa5t93v
> > >
> > > [1] Yinzhu et al. "Learning group actions on latent representations." NuerIPS 2024.
> > >
> > > [2] Yue et al. "Flow factorized representation learning." NuerIPS 2023.
> > >
> > > [3] Jeremiah et al. "Structure-preserving GANs." ICML 2022.
> > >
> > > [4] Tristan et al. "Explicitly disentangling image content from translation and rotation with spatial-VAE." NuerIPS 2019.
> > >
> > > [5] Robin et al. "Unsupervised learning of group invariant and equivariant representations." NuerIPS 2022.
> > >
> > > ---
> > > We thank again the reviewer for these points and we will add the additional demonstration in the manuscript accordingly.

---

### Official Review · Reviewer_WKPv · 2026-03-10

**Soundness:** 3
**Presentation:** 3
**Significance:** 3
**Originality:** 2
**Overall Recommendation:** 4
**Confidence:** 2

**Summary:**

The paper proposes a method for improving learned equivariant representations in encoder-/decoder architectures. The starting point is an autoencoder-type network that creates a latent representation that should act equivariantly under transformations applied to its input (transforming the input should have a homorphic effect on the latent representation). At the baseline, where this paper starts, this effect is obtained by imposing a corresponding loss to the latent encodings (with suitably generated training data that allows for checking violations). The new paper now adds a correction flow that transports embeddings in latent space that are off the intended orbit back to their "proper" (i.e., equivariantly transformed) position. In various experiments, this leads to improvements over the original training objective.

**Compliance With Llm Reviewing Policy:**

Affirmed.

**Final Justification:**

The authors rebuttal addressed my questions, in particular explaining the motivation for the overall problem setting and the generative approach in general (basically working around limitations of the bottleneck-type autoencoder architecture, which seems to me to be a valid reason when committing to this type of approach). I have thus increased my score by one point and also raised my rating for "soundness".

**Key Questions For Authors:**

I have already outlined my main questions in the section above; here is a short summary:

- What is the target application for this method? Why not use an equivariant construction in the first place? If this is impossible, why should it be possible to implicitly enforce it on latents?
- Why use a generative model instead of a direct mapping (maybe neural ODE if one wants to perform incremental corrections)?
- How does the performance of the method sit in the broader spectrum of generic methods (using comparable compute & data, but not necessarily NFT/AEs)?

As a lot of this might be presentation related, I could imagine that clarifications would allow me to better understand the intend and purpose of the method and thus affect my overall recommendation.

**Limitations:**

Social implications seem to be generic, which is mentioned in the paper. I do not miss any additional discussion here. In terms of limitations one could address the aspect of costs of using an encoder-decoder approach in the first place; for some types of problems, this might be a restriction due to information loss in latent space and similar effects.

**Strengths And Weaknesses:**

The paper proposes an interesting addition to the "NFT"-approach of learning symmetry (in the sense of their linear representations) from data. The explicit correction step seems to improve results beyond what the directly trained encoder can achieve. This applies not only to the preservation of symmetry (invariance/covariance) but also to the quality of the results (as this is a trade-off in the original formulation in Eq. 2). I would think that these are the main strengths: The correction step seems to be not to difficult to include and it also seems to improve results.

My perception of the main drawbacks of the submission is centered around two questions, which are also related to how the ideas are presented (so I would assume there is room for clarification/improvement in the discussion phase):

- What is the intended target use case of this method?
- Why use a generative model?

While reading the paper, it seemed to me that an approach that directly constructs an equivariant map (for example: a typical group-equivariant network with steerable linear layers and compatible non-linearities, such as the vast body of work following Cohen et al.'s seminal steerable networks) would solve the same problem in a more principled way. Only in the result sections I realized that the paper is (also?) targeting data where such a construction is not directly possible, such as 2D renderings of objects rotating in 3D. However, in this case, the objects appearance is not solely explained by a linear group action; thus it is not clear in how far the constraint of imposing linear equivariance on the latent embeddings is a good solution. On the contrary, for geometric data (2D and 3D), why not just perform an equivariant construction and it works "perfectly"? My impression is that the motivation for the method and its application domain should be explored/discussed more in depth in the paper to make a point of why the method is useful (significance) and sound.

The second aspect concerns the choice of a generative flow model for remapping the data. This is actually discuss in some detail in the paper, but the arguments do not fully convince me. My understanding is that the paper is looking for an incremental/iterative remapping, as this is more expressive and stable, but does not want the many-to-many aspect of a generative model (which is subsequently removed by conditioning up to delta functions). Why not just use a neural-ODE (a continuous ResNet) to learn a suitable mapping or a fix-point iteration? In the current presentation, the introduction of flow matching seems to overcomplicate the problem in my understanding.

There are a few more aspects to note: A lot of the base-line comparisons are against NFT (and, at some places, other encoder-decoder architectures). It would be useful to put the obtained numbers into context of SOTA-methods not restricted to autoencoder variants (or only the NFT paper). For example, error rates of roughly 10% on ModelNet-10 do not seem very impressive (but again, maybe some context is needed to appreciate the results).

Overall, I am skeptical in terms of the technical approach and the broader motivation of why one should follow it. It did not become clear to me why one would want to use an encoder-with-correction approach (this might be easy to rectify though), why a separate correction step works better than integrating the constraint into the training objective as in Eq. 2, why one should use a generative model as a corrector, and how the results would position themselves in the broader context of methods addressing the same problem (I am not expecting SOTA-results, just a positioning).

For this reason, my overall rating is skeptical, but some of my open questions could probably be rectified during the discussion period.

---

> ### Author Rebuttal · Authors · 2026-03-31
>
> We appreciate the reviewer for the thorough and careful feedback.
>
> ---
>
> __[Q1. Target application and comparison with other equivariant architectures]__
>
> The target application we are focusing in this paper is 3D novel view synthesis (NVS), where we are trying to predict unseen view of a 3D object, under rotation.
> Please see the comment on the Reviewer 1 (Qnzo)'s __[W1. Other equivariant architectures and practical benefits over equivariant NeRF]__.
>
> ---
>
> __[Q2. Why use generative model?]__
>
> A key aspect of our setting is that latent correspondences are not uniquely identifiable in practice. More precisely, the issue arises when there exist distinct inputs $x \neq x'$ such that $\Phi(x) = \Phi(x')$, i.e., the encoder is not injective. In this case, after applying the group action, we obtain two valid targets $\Phi(g \circ x)$ and $\Phi(g \circ x')$ for the same source $\rho(g)\Phi(x) = \rho(g)\Phi(x')$, resulting in a multi-modal target distribution (recall that our source is $\rho(g)\Phi(x)$ and target is $\Phi(gx)$, eq. 5). In natural image data, many different images can share similar latent codes due to compression, invariance, and limited model capacity. Within the ERL framework, this is further exacerbated by imperfect optimization which prevent the encoder from learning perfectly equivariant and injective mappings.
>
> Our method explicitly addresses this regime by framing the problem as learning a transport between distributions rather than a deterministic point-to-point mapping. Flow matching is particularly well-suited here, as it learns a velocity field consistent with the aggregate correspondence structure. We believe this multi-target transport problem, due to latent correspondences being not uniquely identifiable, naturally motivates the use of generative models for latent alignment.
>
> ---
>
> __[Q3. Comparison with different methods for NVS]__
>
> Methods such as [A] by explicitly modeling a 3D-consistent generative process, typically requiring repeated volumetric rendering or integration along camera rays, similar to NeRF. While this yields strong geometric consistency, it comes at a significant computational cost, as each generated view involves iterative diffusion steps combined with heavy rendering operations.
>
> In contrast, our method operates entirely in a compact latent space and learns a direct transport map between representations under group actions. This avoids the need for volumetric rendering or per-ray integration, making the approach substantially more lightweight and efficient. Moreover, our framework is model-agnostic; it can be applied on top of any existing encoders or other generative frameworks that does ERL.
>
>
> [A] Generative Novel View Synthesis with 3D-Aware Diffusion Models
>
> ---
>
> __[Q4. Others ]__
>
> First, regarding the encoder paradigm, this is a light framework for 3D NVS (compared to NeRF or diffusion based methods). Our goal is to explicitly address the gap between ideal equivariance and practical representations under this light weight and model-agnostic (our correction method can be used on any encoder targeting on NVS) framework.
>
> Second, on why not integrate this into the training objective, we emphasize that objective-level constraints alone are often insufficient to eliminate misalignment in practice. Empirically, even strong equivariant regularization does not guarantee exact alignment. In fact, the ERL loss (eq. 2) already has this equivarinace guiding term in the formulation, but still exhibits the latent misalignment.
>
> Finally, in terms of positioning, our method is complementary rather than competing with existing approaches. Architectural methods aim to enforce equivariance by design, while our method corrects residual misalignment when such assumptions do not hold.
>
> ---
> __[L1. Cost of using encoder-decoder]__
>
> We agree that there is an inherent limitation due to the encoder-decoder approach. Since our method operates entirely in latent space, its performance is fundamentally bounded by what the encoder $\Phi$ can represent. In practice, standard autoencoders compress high-dimensional images into compact latents, which inevitably discard high-frequency details necessary for high-resolution synthesis. Moreover, these encoders often entangle geometric transformations with appearance, making it increasingly difficult to enforce consistent equivariance as scene complexity grows, which is the exact problem we are trying to resolve in this paper. Another limitation rises from the computational overheaded by using the encoder-decoder from the first place, as mentioned.
>
> Importantly, we believe our method is fully compatible with stronger latent backbones, such as levaraging SD-VAE in Latent Diffusion Models. We expect that adopting higher-capacity or multi-scale latent representations would directly alleviate the current limitations without requiring changes to the RLF framework itself. We will discuss more about this in the revised manuscript.

---

> > ### Author Rebuttal · Reviewer_WKPv · 2026-04-03
> >
> > Dear authors, thanks for the thoughful replies. This resolves my issues (in particular, taking the answers to R. Qnzo into account, which made the motivation for the generative model much clearer). I will accordingly raise my score.

---

### Official Review · Reviewer_m6g4 · 2026-03-12

**Soundness:** 2
**Presentation:** 3
**Significance:** 2
**Originality:** 3
**Overall Recommendation:** 4
**Confidence:** 3

**Summary:**

This work aims to address an underlying problem of current equivariant representation learning (ERL) methodologies, namely, the misalignment between a latent vector when we apply the group transformation in the input or in the latent vector itself. To address this misalignment, the authors propose to use a flow matching model that learns to align the rotated latent $\rho(g)\Phi(x)$ to the actual latent of the rotated image $\Phi(gx)$. The authors identify one main difference between their proposed distribution alignment and the standard flow matching formulation: the latter assumes independent coupling between the source and target points. In contrast, this work requires a dependence between the source and target, and thus formulates the problem as learning a conditional path where both endpoints are deterministically defined by the pair $(x,g)$. To train the overall model, the authors first train an individual ERL auto-encoder, then use a frozen encoder to train the flow-matching alignment step, and finally fine-tune the decoder using both the frozen encoder and the alignment step. The authors evaluate their proposed framework, showcasing consistent improvements across different datasets and group symmetries (SO(2),SO(3))

**Compliance With Llm Reviewing Policy:**

Affirmed.

**Final Justification:**

The authors' rebuttal have addressed my main questions about the generalization of the proposed framework to different ERL method, and about the ability of the method to handle latent collapse compared to baseline. Due to this I am raising my initial recommendation to weak accept

**Key Questions For Authors:**

- Why is flow matching necessary, and a simple regression model is not sufficient for the current problem, where both endpoints are deterministically chosen?
-  How the proposed methodology address cases where $\rho(g)\Phi(x)=\Phi(x’)$ for $x’\not=gx$. ?
- What is the applicability of the proposed framework in different ERL architectures?

**Limitations:**

Yes

**Strengths And Weaknesses:**

Strengths:

- The problem formulation and the proposed solution are well motivated by an actual limitation of current ERL methods.
- The proposed solution can be generally applied to a large range of models without specific architectural requirements, which boosts the significance of the contribution.
- The experimental evaluation shows significant improvements in results across both synthetic and real-image settings, showcasing that the misalignment module actually benefits the overall model performance.

Weaknesses
- While the authors acknowledge that a regression model would be a natural first approach, they do not provide strong evidence that their choice of flow matching is superior. Either a theoretical or empirical analysis of the differences would strengthen their claim regarding the importance of using a flow-matching model.
- There is an implicit assumption that for  given g,x, there is no $x’\not= gx$ where $\rho(g)\Phi(x)=\Phi(x’)$. Because if such $x’$ exists, it will also be aligned to the incorrect latent $\Phi(gx)$. This assumption is reasonable, assuming that the model optimizes the ERL loss, but in this application, the problem at hand is the actual fact that the models cannot optimize such a loss well during training, leading to misalignment.
- While the presentation of the proposed alignment is architecture agnostic, the authors evaluate mainly  on the NFT architecture. This makes it more difficult to determine if the proposed method is an improvement over a specific NFT failure mode or if it has broader applicability.

---

> ### Author Rebuttal · Authors · 2026-03-31
>
> We appreciate the reviewer for their helpful feedback.
>
> ---
> __[Q1 & Q2. Why generative model over simple regression? How do we address for such x'?]__
>
> Yes, this is one good reason why these latents trained with ERL loss has the inevitable misalignment, due to contraction caused by the imperfect encoder. The issue actually arises when there exist distinct inputs x \neq x' such that $\Phi(x) = \Phi(x')$, i.e., the encoder maps different images to the same latent representation (not $\rho(g)\Phi(x) = \Phi(x')$; We are acting $\rho(g)$ to $\Phi(x')$ before doing the flow correction). In this case, after applying the group action, we obtain two valid targets $\Phi(g \circ x)$ and $\Phi(g \circ x')$ for the same source $\rho(g)\Phi(x) = \rho(g)\Phi(x')$, resulting in a multi-modal target distribution. This is not a pathological case but naturally occurs in realistic settings where we are given an imperfectly trained encoder. In natural image data, different scenes or objects can share similar latent representations due to limited encoder capacity and invariances. Within the ERL framework, this is further exacerbated by imperfect optimization of the equivariance objective, discretization, and architectural biases, which prevent the encoder from learning fully injective and perfectly equivariant mappings.
>
> Our method explicitly addresses this regime by framing the problem as learning a transport between distributions rather than a deterministic point-to-point mapping. Flow matching is particularly well-suited here, as it learns a velocity field consistent with the aggregate correspondence structure. We believe this multi-target transport we are trying to model here, motivated from when latent correspondences are not uniquely identifiable, naturally motivates the use of generative transport models for latent alignment.
>
> One subtle issue in modeling this multi-modal target distribution with an unconditional generative model is the ambiguity in transport: a given latent may be mapped either to $\Phi(gx)$ or $\Phi(gx')$ when $\Phi(x) = \Phi(x')$. In such cases, an unconditional flow model will learn a probabilistic mixture over these possible targets, effectively distributing mass according to the empirical pairing observed during training.
>
> A natural way to resolve this ambiguity is to introduce conditional generation. By conditioning on the input x, the velocity field can be defined as $v_\theta(z_t, t \mid x)$, which guides the latent toward a input-consistent target $\Phi(gx)$. This is analogous to conditioning mechanisms in text-to-image generation or inverse problem solving, where additional context removes ambiguity. We believe incorporating such conditional structure into the flow model could provide a more principled way to handle non-injective latent representations, and we view this as a promising direction for future work.
>
> ---
>
> __[Q3. Generalization beyond NFT]__
>
> To demonstrate that our correction method generalizes across various equivariant representation spaces, we further evaluate it using LGA [A] and observe a substantially reduced latent mismatch. Specifically, the latent error decreases from 10.58 to 0.15 on RotMNIST, and from 1258 to 273 on SmallNORB. Since latent error directly measures the inconsistency between predicted rotated latent and the latent of ground-truth transformed image, these results provide the evidence that our method also effectively corrects the misaligned latents in LGA [A].
>
> [A] Jin, Y., Shrivastava, A., and Fletcher, P. T. Learning group actions on latent representations. Advances in Neural Information Processing Systems, 37:127273–127295, 2024.

---

> > ### Author Rebuttal · Reviewer_m6g4 · 2026-04-03
> >
> > I thank the authors for answering the main questions on my review.
> >
> > I have one clarification question regarding their answer for Q2:
> > They mention that in cases where $\Phi(x)=\Phi(x')$ the unconditional flow model will learn a probabilistic mixture between the two target distributions. But at inference, the learned flow corresponds to integrating a deterministic ODE, so the same initial latent $\rho(g)\Phi(x)=\rho(g)\Phi(x')$ will always be mapped to the same output. Could you clarify how the probabilistic mixture of targets is achieved at inference time?

---

> > > ### Author Response · Authors · 2026-04-05
> > >
> > > Thank you for discussing with a follow-up question!
> > >
> > > Yes, if we input the exact same latent $z$ observed during training, the deterministic ODE will map it to a single target. But at inference time, we are usually interested in generalizing to multiple unseen test data, which does not exactly coincide with the training latent $z$. In fact, every result in our paper presents performance on unseen test data (unseen views); we further test on OOD (unseen objects) data in Tab. 1 and Fig 4, 6, 7.
> > >
> > > In particular, if we consider a finite neighborhood around $z$, the learned flow will effectively partition the region of latent space and transport different inputs to different target modes learned during training. It is reasonable to expect that images $y_i$ with visual features similar to $x$ and $x'$, whose corresponding latents $\Phi(y_i)$ lie close to $z = z'$ (but not exactly same with $z$), will each follow different trajectories under the trained flow and ultimately map to different target modes, such as to the neighborhood of $\Phi(\rho(g)z)$ or that of $\Phi(\rho(g)z')$. Hence, we want to clarify that the probabilistic mixture is achieved at the distribution level at testing time.
> > >
> > > We thank again the reviewer for leading us to the discussion. We will revise the expression and description of this argument in the manuscript for clarity.

---

### Official Review · Reviewer_Qnzo · 2026-03-13

**Soundness:** 3
**Presentation:** 3
**Significance:** 2
**Originality:** 2
**Overall Recommendation:** 4
**Confidence:** 2

**Summary:**

The authors propose Residual Latent Flow, a flow matching module trained to transport the analytically rotated latent toward encoded latent. The method treats analytically transformed latents as imperfect approximations and learns a residual flow to align them with empirically encoded representations. The empirical evaluation consisted of  six datasets covering SO(2) and SO(3) symmetries, demonstrating consistent improvements in metrics like PSNR, prediction error, latent error, etc.

**Compliance With Llm Reviewing Policy:**

Affirmed.

**Final Justification:**

The authors responded to my concerns. I am not fully convinced about a few things like purely linearly interpolation working better than stochastic as well as the authors response on equivariant VAEs mentioned in the review.
Although these points do not affect the quality of the work. Thus, I have increased the score but reduced my confidence.

**Key Questions For Authors:**

1. Linear interpolation is considered in the paper. Any comment on stochastic interpolation?

2. For evaluation of misalignment are higher-order components considered?

3. Is it possible to comment on generalized beyond NFT? With just a single architecture design it is difficult to understand the true working of the approach.

4. As far as I understand, the framework requires pairs for training. Suppose that isn't the case or some are miss- paired, how does this method work? Is there some uncertainty that could be defined over the miss pairing?

5. Is this approach easily modified for various groups? How about SE(3)?

**Limitations:**

Yes

**Strengths And Weaknesses:**

## Strengths
1. The paper is well written and easily readable for the most parts.
2. A posthoc correction algorithm for latents in an equivariant model to have a better equivariance check is an interesting idea.
3. The figures are good and OOD experiments are a good addition, as they are usually left out.



## Weaknesses
1. The scope of the work feels a bit limited, though not because of scope of the problem tackled. But that a few important points are still not fully understood; Architectural affects to this approach, Equivariant architecture that can input or extract pose (or rotation angles in this case [1,3]) can prevent misalignment so what is the true benefit of the approach in that case. Also, what is the practical benefit of this approach vs equivariant NeRF?
2. It would be good to see a discussion about NeRFs or equivariant neural fields [2], as they do have this issue and would be interesting to see authors comments on alternate approaches.
3. Works that do not face with representational alignment issues are not cited. Equivariant VAEs  [3,4]


### Citations

[1]Fast, Expressive  Equivariant Networks through Weight-Sharing in Position-Orientation Space, Bekkers et al


[2] Grounding Continuous Representations in Geometry: Equivariant Neural Fields, Wessels et al

[3] Kendal Shape VAEs, Vadgama et al

[4] Continuous kendall shape variational autoencoders, Vadgama et al

---

> ### Author Rebuttal · Authors · 2026-03-31
>
> We appreciate the reviewer for valuable feedback.
>
> ---
>
> __[W1. Other equivariant architectures and practical benefits over equivariant NeRF]__
>
> We clarify these works focus on enforcing equivariance at the representation or architectural level, often by explicitly encoding pose variables or constructing features that transform according to a predefined group action. These works do not directly address the 3D novel view synthesis (NVS) considered in our work, where one must infer consistent representations across unseen viewpoints. For example, [1], [3], and [4] addresses in-plane rotation problem, which does not require inferring about occluded views of a higher-dimensional object. In our paper, we try to solve NVS with out-of-plane rotation and we believe extending these equivariant architecture, to this particular problem is not straightforward. Similarly, Cohen et al. Steerable CNNs or Group equivariant CNNs explicitly constructs of equivariance relation directly on the data itself, while the novel view synthesis task we are focusing in this paper is about inferring unseen data, given only a partial view of an 3D object.
>
> In practice, such latents entangle appearance and geometry, and even with equivariant or pose-aware designs, approximate equivariance errors persist due to discretization and optimization limitations. While these works enforce equivariance by design; we fix it when that design fails in practice.
>
> Compared to Equivariant Neural Fields, our method offers a practical advantage in that it operates in a compact latent space without requiring continuous field representations, dense multi-view supervision, or modifications to the rendering pipeline. NeRF requires heavy integration of the continuous volumetric field and this makes our approach significantly more lightweight and also directly compatible with different types of generative frameworks (e.g., autoencoder + diffusion). In this sense, our method provides a general correction mechanism that enhances equivariance in realistic NVS pipelines where ideal architectural assumptions do not hold.
>
> ---
>
> __[Q1. About stochastic interpolation]__
>
> In Appendix. H, we discuss about the stochastic interpolations. We believed stochastic paths could provide some advantages, but we noticed linear path without noise was the most effective interpolation to be used during training.
>
> ---
>
> __[Q2. Evaluation on higher-order components]__
>
> Yes, we use latent errors to quantify the error in higher-order components (for all degree \ell-vectors). Latent errors are defined as described in Section. 4.2 and the results can be seen in Table. 1.
>
> ---
>
> __[Q3. Generalization beyond NFT]__
>
> Yes, our flow-based correction is applicable to other ERL frameoworks, beyon NFT. Please see the comment on the Reviewer 2(m6g4)'s __[Q3. Generalization beyond NFT]__.
>
> ---
>
> __[Q4. Uncertainty in the train data pairing?]__
>
> Yes, there could be an uncertainty in the pairing and this could be an interesting question to address. Our current formulation indeed assumes access to paired latents (\rho(g)\Phi(x), \Phi(g \circ x)) during training. However, the framework is inherently robust to moderate mispairing, as the flow is learned over a distribution of correspondences and effectively averages consistent transport directions; incorrect pairs behave as noise and will not induce a coherent vector field; however if the noise level is too large, it will eventually impair the training.
>
> Importantly, the flow-based correction method admits a natural extension to uncertain or unpaired settings. Instead of assuming a deterministic target, one can model a conditional distribution over the targets with some uncertainty and learn the flow under a soft matching objective, e.g., by weighting candidate pairs with the given pairings. In this view, mispairing can be interpreted as uncertainty over the coupling between source and target distributions, and the learned flow will implicitly reflects this uncertainty.
>
> We leave this extension as future work, but emphasize that the proposed formulation is not fundamentally limited to perfectly paired data and can be generalized to settings with ambiguous or partially observed correspondences.
>
> ---
>
> __[Q5. Extension to SE(3)]__
>
> Yes, our residual latent flow is utilizing flow matching to learn the residual term in the representation of group elements, and this is extensible to SE(3). Particularly, since SE(3) is a semi-direct product of translation group and SO(3), we could use fourier representation and wigner D representation for each part. This will require restricting the range of translation to some finite number, since there are infinite number of irreducible representation for non-compact group, such as SE(3). Hence, learning a residual flow using this representation with our method would be an effective way to improve equivariance of the learned representations.

---

> > ### Author Rebuttal · Reviewer_Qnzo · 2026-04-03
> >
> > I do not have further questions for the authors.

---

### Decision · Program_Chairs · 2026-04-30

**Decision:**

Accept (regular)

**Comment:**

All reviewers agree that the paper considers a real and important failure mode in equivariant representation learning (latent misalignment under group actions) and proposes a (flow-matching) correction. The reviewers agree that the paper is well-written, well-motivated, and technically solid, with clear figures and broad experimental experiments.

The main concerns are related to the motivation behind the proposed method (why flow-matching? why not other (simpler) methods? why evaluate on the NFT architecture?) The rebuttal largely resolves the reviewers' concerns.

After the rebuttal, all reviewers converged to weak accept.